# Destructiveness of pyroclastic surges controlled by turbulent fluctuations

Ermanno Brosch [1✉], Gert Lube [1], Matteo Cerminara [2], Tomaso Esposti-Ongaro [2], Eric C. P. Breard [3], Josef Dufek[3], Betty Sovilla [4] & Luke Fullard[5]

Pyroclastic surges are lethal hazards from volcanoes that exhibit enormous destructiveness through dynamic pressures of $10^0$–$10^2$ kPa inside flows capable of obliterating reinforced buildings. However, to date, there are no measurements inside these currents to quantify the dynamics of this important hazard process. Here we show, through large-scale experiments and the first field measurement of pressure inside pyroclastic surges, that dynamic pressure energy is mostly carried by large-scale coherent turbulent structures and gravity waves. These perpetuate as low-frequency high-pressure pulses downcurrent, form maxima in the flow energy spectra and drive a turbulent energy cascade. The pressure maxima exceed mean values, which are traditionally estimated for hazard assessments, manifold. The frequency of the most energetic coherent turbulent structures is bounded by a critical Strouhal number of ~0.3, allowing quantitative predictions. This explains the destructiveness of real-world flows through the development of c. 1–20 successive high-pressure pulses per minute. This discovery, which is also applicable to powder snow avalanches, necessitates a re-evaluation of hazard models that aim to forecast and mitigate volcanic hazard impacts globally.

[1] Volcanic Risk Solutions, Massey University, Palmerston North, New Zealand. [2] Istituto Nazionale di Geofisica e Vulcanologia–Sezione di Pisa, Pisa, Italy. [3] Department of Earth Sciences, University of Oregon, Eugene, USA. [4] WSL Institute for Snow and Avalanche Research SLF, Davos, Switzerland. [5] School of Fundamental Sciences, Massey University, Palmerston North, New Zealand. ✉email: e.brosch@massey.ac.nz

The destruction potential of pyroclastic surges (also called dilute pyroclastic density currents) is a critical and outstanding uncertainty in volcanic hazard studies[1–3]. Over 100 million people worldwide are potentially endangered by these fast moving (10s to 100s of m s$^{-1}$), fully turbulent and ground-hugging gravity currents of hot volcanic particles and gas[4,5]. Their significant threat to life and their ability to catastrophically damage reinforced buildings, infrastructure[6–9] and forests[10] results from internal dynamic pressures of 10s to 100s of kPa and remains poorly mitigated globally. Thus, the development of robust hazard models that can explain and predict how large dynamic pressures manifest or persist over long flow runouts and across significant topographic obstacles is a priority in volcanology and natural hazard science[11]. But the violence of pyroclastic density currents has precluded direct measurements of their inner workings, so that the mechanisms leading to such levels of destruction have never been directly observed or measured. In fact, our knowledge of the highly destructive dynamic pressures inside flows derives from broad estimates of the degree of damage that can be observed after an eruption[12].

Over the past two decades, and to address the urgent need for volcanic hazard assessments globally, strategies have been adopted to determine transport parameters of pyroclastic surges from their deposits. For instance, local profiles of the average dynamic pressure of past eruptions can be estimated by methods that combine principles of the turbulent-boundary-layer theory and sediment transport hydraulics[13,14]. Furthermore, the local mean flow velocity has been estimated from deposit characteristics through an approximation of the conditions of gas-particle decoupling and turbulent sedimentation from large eddies[15,16]. Volcanologists can capitalize on the wealth of pyroclastic surge deposits and use these estimates to reconstruct local time-averaged conditions of past eruptions to inform hazard assessments for future events[17].

However, the turbulence structure of pyroclastic surges is still poorly constrained. Fundamental gaps in understanding remain on how gas-particle turbulence, thermal buoyancy, polydispersity (large particle size distribution of the mixture) and wall shear on natural surfaces modify pressure profiles, energy dissipation, flow stratification and particle settling inside pyroclastic surges[11]. This hampers hazard model application due to the difficulty of testing multiphase turbulence closure schemes[11,18] and it leaves critical model assumptions such as decaying isotropic turbulence (that is the idealistic state of turbulence, where turbulent fluctuations are assumed to decay statistically uniformly in every direction) unvalidated. Until we can explain the complex gas-particle transport and feedback mechanisms behind the extreme ferocity of pyroclastic surges (and pyroclastic density currents in general), we cannot adequately forecast hazard impacts for the millions of people at risk[2,19].

In this article, through large-scale experiments, we clarify the characteristics and perpetuation mechanism of destruction-causing dynamic pressure in pyroclastic surges. Furthermore, we present the first measurements of pressure inside natural pyroclastic density currents and compare these with the results of the experiment. Pyroclastic surges generated during the 9 December 2019 eruption of Whakaari (White Island) killed 22 visitors to the island and severely injured another 25 marking this the deadliest eruption in New Zealand since the 1886 eruption of Mount Tarawera[20]. An intriguing aspect about the characteristically low-intensity phreatic eruption is the overrunning of the surges of an infrasound monitoring array capturing a time series of pressure inside the flow. These direct measurements inside PDCs are distinct from previous geophysical signals induced by PDCs in the ground and atmosphere surrounding the flow e.g.[21–23]. Such seismic, acoustic and radar signals currently require interpretation based on models of the PDC structure and interaction with the environment to infer aspects of the flow dynamics. In both the large-scale experiments and in the unique real-world measurements, we demonstrate the role of coherent turbulence structures in exacerbating hazard magnitudes. Through this, we refine our understanding of hazard impacts of pyroclastic surges and other high-energy gas-particle gravity currents that necessitates a re-evaluation of current hazard assessment strategies.

## Results

**Synthesizing pyroclastic surges in large-scale experiments**. To study the generation of high dynamic pressure inside pyroclastic surges we synthesized them in large-scale experiments using the international eruption simulation facility PELE in New Zealand (Pyroclastic flow Eruption Large-scale Experiment)[24]. This extends previous research reported to date at other large-scale and medium-scale facilities in Italy[25], the USA[26] and Mexico[27], as well as earlier work at PELE on pyroclastic flows[28–30]. At PELE, experimental pyroclastic surges are generated by the controlled gravitational collapse of a hot, aerated suspension of natural volcanic particles and air from an elevated hopper into an instrumented runout section. For the experiments reported here, we utilize a 0.7 m$^3$ hopper, which heats a 124 kg mixture of natural volcanic particles to an initial temperature of 120 °C (the ambient temperature was 11 °C) over a period of three days to allow thermal equilibration and evaporation of moisture inside the pre-dried mixture. Supplementary Table 1 summarises the initial and boundary conditions of the experiment.

The volcanic material consists of a mixture of two well-characterised pyroclastic density current deposits of the 232 CE Taupo eruption in New Zealand[31]. The mixture contains highly vesicular pumice, glass shards, free crystals and rare lithic particles; it has a weakly bimodal grain-size distribution ranging from 2 μm to 16 mm with a main mode at 250 μm and a minor mode at 11 μm. The content of very fine ash (particles <63 μm) is approximately 20 wt.%. Further details of the material characteristics are provided in Supplementary Fig. 1a, b and in the Methods section.

The hopper is lifted inside a 13-m high vertical elevator structure to a vertical drop height of 7 m. The hopper is mounted onto four load cells recording its time-variant mass discharge. Horizontal diamond-shaped bars regulate the hopper mass discharge, which in this case lasts for 5.2 s with a unimodal discharge rate that is characterised by a peak value of 44 kg s$^{-1}$ at half-discharge time and a time-averaged value of 24 kg s$^{-1}$ (Supplementary Fig. 2). The mixture falls into a 12-metre long, 0.5-metre wide and 6° inclined channel before spreading out onto a flat concrete pad (Fig. 1a). Sub-rounded rock pebbles (4–8 mm in diameter) were glued to the channel base, generating an effective substrate roughness of 5 mm. This simulates, for the case of pyroclastic density currents with thicknesses of 50–500 m, a scaled c. 0.1–1 m-rough non-erodible volcanic surface.

On impact with the channel, at a vertical velocity of c. 7 m s$^{-1}$, the suspension simulates a directed pyroclastic density current and initially contains, on average, c. 0.24 vol.% solids (that is a solid mass fraction $\Phi_M$ of 0.79). Approximately half a second after impact (at 3.12 m runout distance), the mixture starts to form a typical gravity current structure. This is characterised by a c. 1–2.5 m thick gravity current head, and a trailing, c. 1–1.6 m thick gravity current body, which is overlain by the gravity current wake (Fig. 1b). At this runout distance, the bulk particle concentration has further decreased by ambient air entrainment and sedimentation to c. 0.11 vol.% ($\Phi_M = 0.63$). This translates to a bulk flow density of c. 3.4 kg m$^{-3}$ and a density ratio with the ambient air of approximately three.

On moving down the flume (Supplementary Movie 1), the flow becomes vertically stratified with regards to particle solids-concentration forming an upper, 1.5–2.5 m thick, fully dilute (<0.2 vol.%) fully turbulent region, above a basal. c. 0.01–0.08 m thick concentrated bedload region[32]. During runout, the flow front velocity generally decreases from c. 7 m s$^{-1}$ at impact to <0.1 m s$^{-1}$ beyond 30 m runout. Between 4 and 7 s after impact,

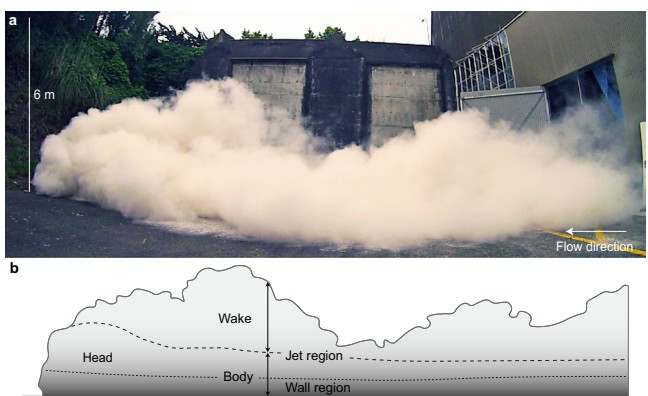

**Fig. 1 Synthesizing pyroclastic surges in large-scale experiments. a** lateral view of an advancing experimental pyroclastic surge at the eruption simulator PELE. **b** Schematic sketch of the internal flow structure shown in Fig. 1a. This illustrates the longitudinal subdivision of the gravity current into the frontal head and trailing body regions; and the vertical structure of the gravity current where the rear of the head and the entire body region are overlain by the gravity current wake. The wide dashed line demarcates the position of the lower boundary of the wake. The narrow-dashed line delineates the position of the fastest velocity magnitude in vertical profiles through the current. This position separates the head and body regions into a lower layer dominated by shear with the solid lower flow boundary, which is often referred to as the wall region; and an upper region dominated by free shear, which is often termed the jet region. Figure 1 depicts the experimental surge during propagation across the unconfined runout section from c. 16–31 m from the impact zone. At this stage, partially buoyant plumes develop across the flow length, which continue rising as a phoenix cloud for tens of seconds after the cessation of forward motion of the experimental pyroclastic surge.

an upper part of the current rises buoyantly along its entire runout length to heights between 8 and 20 m. This 'phoenix cloud' deposits its very fine ash load over a duration of several minutes, while emplacement of the laterally moving and coarser-grained main part of the experimental pyroclastic surge occurs in less than 25 s. Deposits from these experiments are c. 0.001–0.2 m thick.

Six vertical arrays with a total of 200 sensors are positioned at runout distances of 1.9, 3.12, 5.77, 10.9, 14.5, 17.8 and 21.5 m to measure time-variant velocity, particle concentration and temperature profiles of the propagating current (see ref. [32] and Methods for details). Flow velocity components $u(z, t)$ are measured using high-speed video through the flume's temperature-resistant glass walls. These sidewalls introduce boundary effects that are not present in unconfined real-world surges. In our experiments, we minimize these effects through the use of hydraulically smooth sidewalls (laminar layer thickness/wall roughness <5) and through the flow's high-Reynolds number (Re = $1.5 \times 10^6$), which is inversely related to the thickness of the viscous boundary layer. Sensors are positioned at a distance of several centimetres from sidewalls to protrude through boundary layers. Vertical sensor profiles measure the time (t)-variant and height (z)-variant grain-size distribution, volumetric particle concentration $C_S(z, t)$ and temperatures $T(z, t)$, from which we calculate dynamic pressure inside the flows, defined as

$$P_{\text{dyn}} = \frac{1}{2} \rho_C |u|^2 \tag{1}$$

where $z$ is the height in slope-perpendicular direction, and $\rho_C(z, t)$ is the local flow density of the current and $|u|$ (z, t) the magnitude of the local flow velocity.

To demonstrate scaling to real-world flows, Table 1 compares non-dimensional products of the characteristic length-, time-, velocity- and temperature scales of the experimental and natural pyroclastic surges. There is a good match as exemplified by the Reynolds numbers (a measure of turbulence intensity) reaching values of $1.5 \times 10^6$, Richardson numbers (characterising the stratification stability in turbulent flows) of 0.01–10, thermal Richardson numbers (assessing the ratio of forced and buoyant convection) of 0.02–4.5, Stokes numbers (characterising particle coupling to turbulent flow) of $10^{-3}$–$10^0$, and Stability numbers

**Table 1 Comparison of bulk flow scaling of natural pyroclastic surges[63-65] and experimental pyroclastic surges in PELE large-scale experiments.**

| Parameter | Formula | PDCs PELE | PDCs nature |
|---|---|---|---|
| Particle diameter | | $10^{-6}$–$10^{-2}$ m | $10^{-6}$–$10^{-1}$ m |
| Solids density | | 350–2600 kg m$^{-3}$ | 300–2600 kg m$^{-3}$ |
| Ambient density | | 0.8–1.2 kg m$^{-3}$ | 0.6–1.2 kg m$^{-3}$ |
| Ambient dynamic viscosity | | $3 \times 10^{-5}$–$3 \times 10^{-3}$ kg m$^{-1}$ s$^{-1}$ | $1 \times 10^{-5}$–$4 \times 10^{-3}$ kg m$^{-1}$ s$^{-1}$ |
| Typical velocity | | <0.5–9 m s$^{-1}$ | 10–200 m s$^{-1}$ |
| Kinetic energy density | | $10^{-2}$–$10^3$ J m$^{-3}$ | $10^3$–$10^4$ J m$^{-3}$ |
| Buoyant thermal energy density | | $10^1$–$10^3$ J m$^{-3}$ | $10^3$–$10^4$ J m$^{-3}$ |
| Reynolds number | $\frac{\rho_c U h}{\mu_c}$ | $4.8 \times 10^4$–$1.5 \times 10^6$ | $3.3 \times 10^6$–$6.7 \times 10^9$ |
| Richardson number | $\frac{\triangle \rho h g}{\rho_a U^2}$ | 0.01–10 | 0–10 |
| Thermal Richardson number | $\frac{\triangle T \alpha h g}{U^2}$ | 0.02–4.5 | 0–5 |
| Froude number | $\frac{U}{\sqrt{g' h \cos(\theta)}}$ | 0.75–2 | c. 1 |
| Stokes number | $\frac{U_T \triangle U_i}{\delta g}$ | $1 \times 10^{-3}$–$9.9 \times 10^0$ | $1.1 \times 10^{-3}$–$9.7 \times 10^7$ |
| Stability number | $\frac{U_T}{\triangle U_i}$ | $1.3 \times 10^{-2}$–$3.2 \times 10^1$ | $2.8 \times 10^{-6}$–$9.7 \times 10^9$ |
| Rouse number | $\frac{U_T}{k U_s}$ | $6.6 \times 10^{-1}$–$1.9 \times 10^1$ | $10^{-3}$–$10^2$ |

Bulk flow scaling is conducted with the Reynolds number, the Richardson number, the thermal Richardson number, the Froude number, the Stokes number, the Stability number and the Rouse number. $h$ is the flow height; $\rho_c$ is flow density, $\rho_a$ is ambient density and $\Delta\rho$ is the difference between flow and ambient densities; $\Delta T$ is the temperature difference between the flow and ambient air; $U_T$ is the terminal fall velocity of solid particles; $\Delta U_i$ is the eddy rotation velocity; $U_s$ is the shear velocity; $\delta$ is the eddy diameter; $\alpha$ is the thermal expansion coefficient of air; $\mu_c$ is the dynamic viscosity of the flow, $g$ is gravity, $g'$ is the reduced gravity, $k$ is the von Karman constant and $\Theta$ is the slope of the substrate.

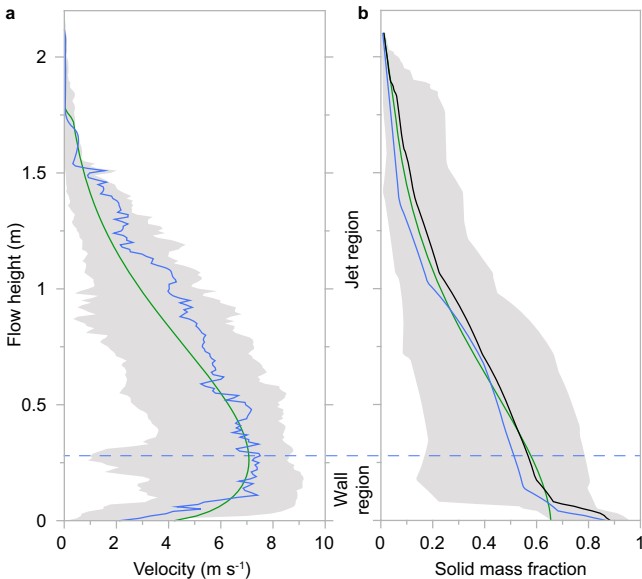

**Fig. 2 Form and time-variance of vertical profiles of velocity and solid mass fraction. a** typical example of the magnitude of flow velocity as a function of height (solid blue line) as obtained by particle image velocimetry at a static observer location of 3.12 m. The horizontal dashed blue line outlines the vertical position of the instantaneous velocity maximum that delineates the boundary between the wall and jet regions. The green line shows the fit of the power-Gaussian velocity model (Eq. 2) to these data. The grey shaded area outlines the total temporal fluctuations of the velocity magnitude as a function of height. **b** typical example of the flow's solid mass fraction as a function of height (solid blue line) at a given time at a static observer location of 3.12 m. The green line shows the fit to the measured data by the empirical model of ref. [34] developed for dilute turbidity currents, which is defined as $\hat{\gamma}_S(\eta) = \vartheta \exp(\kappa \eta^\nu)$, where $\gamma_S$ is solid mass fraction, $\eta$ is non-dimensional height, and $\vartheta$, $\kappa$ and $\nu$ are free fitting parameters. In the wall region, the empirical model deviates from the experimental data due to fast settling mesoscale turbulence clusters that occur in our experimental pyroclastic surges[11,28,32], but not in dilute turbidity currents. The time-integrated vertical profile of solid mass fraction is presented as the black line. The grey shaded area outlines the total temporal fluctuation in solid mass fraction as a function of height at the static observer location.

(comparing velocities of particle settling relative to turbulent fluid motion) of $10^{-2}$–$10^1$. The experimental ranges in Reynolds, Stokes and Stability numbers, together, ensure that the complete range of gas-particle feedback mechanisms and turbulent particle transport in eddies is realized.

**Self-generated pulsing in pyroclastic surges.** To date, hazard models for pyroclastic surges have large uncertainties due to a lack of analytical models that define the form and downstream evolution of vertical profiles of velocity and density (or particle concentration) and their possible interdependence.

To visualise the internal flow structure, we measure high-resolution height- and time-variant velocity fields at a single location. The high-Reynolds number current (Re = $1.5 \times 10^6$) develops a classic boundary layer time-averaged velocity profile with an inner wall region and an outer jet region, which is related to the generation of shear at the lower solid and upper free-shear flow boundaries (Fig. 2a). This characteristic form of vertical velocity profiles is well-fitted by earlier empirical models developed for dilute turbidity currents[33,34]. Integrating these studies on particle-laden gravity currents with recent studies on volcanic plumes[35,36], we here propose a continuous power-

Gaussian mathematical form of the mean vertical velocity profile as

$$u(\eta) = U_m \eta^\xi \exp\left[-\left(\frac{\eta - 1}{\chi}\right)^2 - \xi(\eta - 1)\right] \qquad (2)$$

where the dimensionless height $\eta = z/h_m(t)$, with $h_m$ being the height of the wall region where the velocity maximum $U_m$ occurs; $\xi(t)$ is the wall layer exponent; $\chi(t)$ is the jet region exponent; and where $U_m$, $h_m$, $\xi$ and $\chi$ are fitted variables. Equation (2) is a differentiable version of the profile proposed by ref. [34] who proposed a power law in the boundary layer and a Gaussian profile in the outer layer.

Vertical profiles of solid mass fraction demonstrate the strong vertical density stratification inside the experimental surges (Fig. 2b). In the jet region, the density profiles are well described by current empirical models developed for dilute gravity currents[34]. However, in the wall region, vertical density gradients are strongly enhanced due to the fast settling of mesoscale turbulent particle clusters[32,37,38] and neither instantaneous nor time-integrated concentration profiles are well described by existing empirical models[34].

The temporal evolution of the vertical velocity and flow density structures is illustrated in height vs. time contour plots in Fig. 3a, b. Markedly, the velocity data are characterised by the occurrence of regular long-period oscillations that occur at intervals of approximately 800 ms. The passage, at our local observer location, of four to five velocity peaks in 4 s is 'mimicked' by large-scale surface waves at the upper flow boundary. Moreover, a similarly regular pattern of oscillation at c. 800 ms also occurs in the data of time-variant flow density (Fig. 3b).

The velocity and density time-series data can be combined to compute the dynamic pressure field for the current passing the static observer location (Eq. 1; Fig. 3c). Due to the temporal correlation of the velocity and density oscillations, the time-variant dynamic pressure is also characterised by the passage of marked pressure oscillations that show the same regular period of c. 800 ms. Importantly, dynamic pressure remains high throughout the flow passage with maximum values of several tens to hundreds of Pascals occurring in the wall region. The measurements of dynamic pressure oscillations are surprising because our conceptual models of dilute pyroclastic density currents, which are based on an analogy to moderately turbulent aqueous particle-laden density currents, envisage a single pressure peak associated with the passage of the current's head[39,40]. However, the occurrence of flow-internal pulses has been envisaged in sedimentological studies e.g.[41–43].

The velocity data lends itself to track the low-frequency oscillatory pattern downcurrent (Fig. 4). This shows that the oscillations persist during flow runout and advance through the propagating current. Our data shows that over the initial 18 m of runout the period of oscillation only increases slightly from c. 800 to 1000 ms. This is associated with a downstream decay of the streamwise length scale of the oscillations from c. 3.2 to 1.6 m, while the integral vertical scale (that is the height of the flow body) remains relatively constant with an average thickness of 1.2 m.

We also conducted experiments with the same starting conditions but a larger initial mass of 300 kg that resulted in longer hopper discharge times. In this situation, the duration of flow passage increases, and eight to nine oscillations occur over 7 s yielding a similar oscillation frequency as the experimental run with initial mass of 124 kg.

The phenomenon of a pulsating velocity structure of gravity currents has been recently recognized in field measurements,

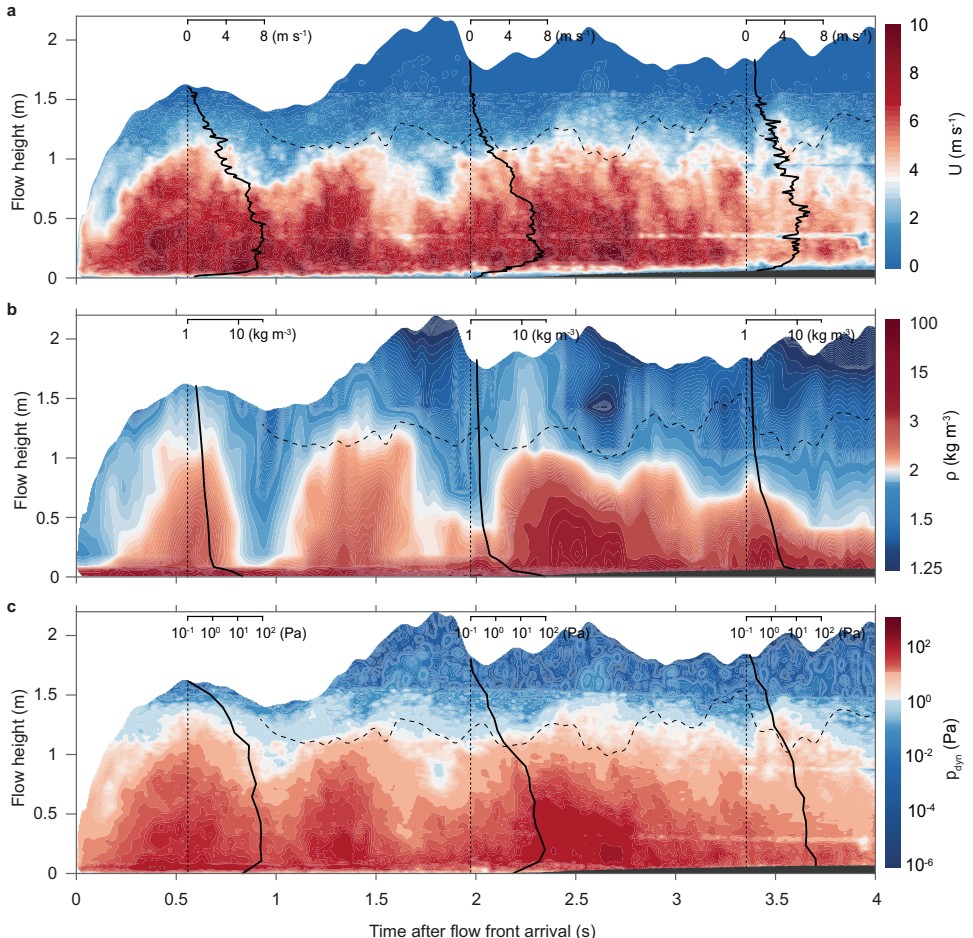

**Fig. 3 Low-frequency oscillations of flow velocity, density and dynamic pressure and their temporal correlation.** Time- and height-variant evolution of the internal flow structure at a static observer location at 3.12 m. **a** contour plot of velocity magnitude $U$ and three instantaneous vertical profiles of velocity at 550, 1980 and 3350 ms after flow arrival. The data shows regular oscillations of velocity at intervals of c. 800 ms. **b** contour plot of flow density fields $\rho$ and three instantaneous vertical profiles of density at 550, 1980 and 3350 ms after flow arrival. The data also shows regular oscillations in flow density at intervals of c. 800 ms. **c**, contour plot of dynamic pressure $p_{dyn}$ and three instantaneous vertical profiles of dynamic pressure at 550, 1980 and 3350 ms after flow arrival. The pressure data also shows the regular oscillations at intervals of c. 800 ms that occur in the velocity and density data. High dynamic pressures of 10s to 100s of Pa persist during the entire flow passage. Dashed lines in contour plots mark the boundary between the gravity current body and wake regions. The aggrading deposit at the base of the flow is in dark grey.

laboratory experiments and numerical simulations of continuous turbidity currents[44–46]. In our experiments, the generation and downstream perpetuation of regular velocity, concentration and dynamic pressure oscillations occurs despite a unimodal source mass discharge rate and appears less than a second after impact and thus concurrent with the formation of a gravity current from the collapsing suspension.

In addition to the strongly time-correlated velocity, density and dynamic pressure oscillations, we observed the occurrence of regular density discontinuities (or pulses) that travel downstream through the advancing density current. These discontinuities in flow density are clearly visible and audible to a static observer standing next to the flume sidewalls. They appear as fast-travelling bow-shaped structures that are most dominant in the wall region and the lower jet region and they have visibly higher flow density than immediately preceding and following flow regions (inset in Fig. 5). We detected five individual pulses that occur at an average period of c. 750 ms and tracked their passage through the experimental surge (Fig. 5). The average downstream velocity of the pulses $c = 6.72 \, \text{m s}^{-1}$ is considerably faster than the surge front velocity and results in the discontinuities catching up subsequently with the head.

**The engines of turbulence generation**. We hypothesize that the marked oscillations in velocity and concentration are the result of the development of coherent, large-eddy structures in the turbulent flow. To visualize the turbulence structure of our experimental pyroclastic density currents, we decompose the height- and time-variant velocity data of the flow passing the static observer location into their mean and fluctuating parts. Turbulent fluctuations for both the downstream and orthogonal velocity components $u'$ and $v'$ are computed following

$$u'(z, t) = u(z, t) - u_{model}(z, t) \qquad (3)$$

$$v'(z, t) = v(z, t) - \overline{v_{model}}(z) \qquad (4)$$

as the difference between the measured (raw) velocity data of both components $u$ and $v$ obtained from particle image velocimetry and velocity integrates. We obtain the model velocity $u_{model}(z, t)$ through fitting the free parameters in the power-Gaussian model (Eq. 2) in time and height by applying an optimising algorithm to find the optimum fit of the mean velocity profile of Eq. (2) as a function of time (Supplementary Movie 2 and Methods section). The model velocity $\overline{v_{model}}$ is computed as the time-integrated vertical velocity component.

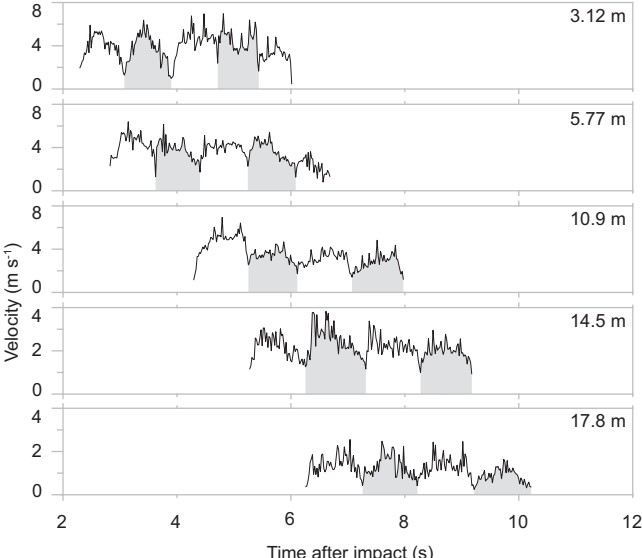

**Fig. 4 Spatial evolution and downstream propagation of low-frequency flow oscillations.** Time-series data of velocity magnitude against time in the middle of the jet region (at a flow height of 0.8 m) at static observer locations situated at runout distances of 3.12 m, 5.77 m, 10.9 m, 14.5 m, and 17.8 m. The regular low-frequency oscillation pattern can be tracked downstream. White and grey shaded areas outline the approximate periods of oscillation. The duration of these periods increases only slightly during flow runout from c. 800–1000 ms and have an average duration of 870 ms.

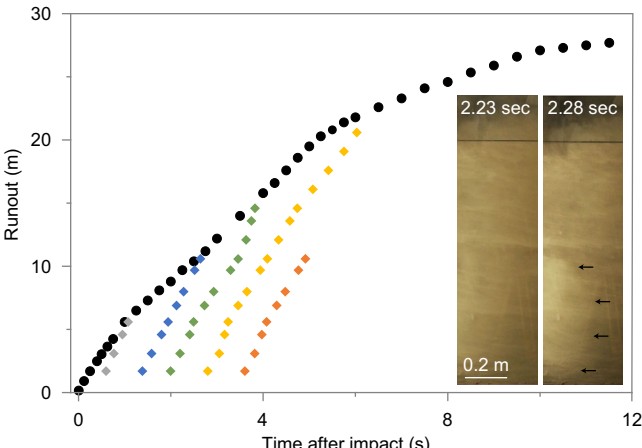

**Fig. 5 Kinematics of the surge front and flow density discontinuities (pulses).** Position of the flow front as a function of time (black dots) from flow impact to a runout distance of 28 m (datapoints for the last four meters of propagation are not shown). The positions of the fronts of the five fast-travelling internal density discontinuities (pulses) as a function of time are shown as grey, blue, green, yellow and orange diamond symbols. The inset depicts two high-speed images of the lower 1.3 m of the flow at different time (2.23 and 2.28 s after impact). The image at 2.28 s shows the passage of a bow-shaped internal density discontinuity whose front is highlighted by black arrows. The feature stretches vertically across the entire wall region and the lower third of the jet region.

The resulting field of turbulent fluctuations reveals the coherent turbulence structures of the experimental pyroclastic surge (Fig. 6a). Shear with the rough bed, creating the turbulent-boundary layer, and free shear of the turbulent flow with the unconfined atmosphere generate the largest (0.9–1.2 m) eddies that pass at a period of c. 800 ms.

The high temporal resolution of the velocity data lends itself to the computation of the energy spectra of the specific kinetic energy $\widehat{\varepsilon_k}$ for both the wall and the jet region, defined as

$$\widehat{\varepsilon_k} = \frac{1}{2}[(u'(\tilde{z},t))^2 + (v'(\tilde{z},t))^2] \quad (5)$$

The distributions of specific kinetic energy in both regions in Fourier (frequency) space are similar (Fig. 6b). Their spectra display the classic Kolmogorov energy cascade (with slope equal to −5/3) of the turbulent inertial range[47,48] up to c. 50 Hz, above which the white noise of the velocity measurements overcomes the fluctuation amplitude. The largest eddies, generated by shear between the top of the current and atmosphere, form the high-energy start of the spectra and energy cascade. In the jet region, the frequency of the largest eddies carrying the greatest energies determined in this way peaks at 1.25 Hz (Fig. 6b). In the wall region, the 1.25 Hz frequency forms the second largest energy, while the peak of the energy maximum is situated at 1.75 Hz.

**Turbulence-enforced destruction**. Considering PDC hazard impacts, it is interesting to determine the energy spectra of dynamic pressure $\hat{P}_{\text{dyn}}$, defined as

$$\hat{P}_{\text{dyn}} = \frac{1}{2}\left[\left(\sqrt{\widetilde{\rho_C(z,t)}}u'(z,t)\right)^2 + \left(\sqrt{\widetilde{\rho_C(z,t)}}v'(z,t)\right)^2\right] \quad (6)$$

In the jet region, and similar to the specific energy, dynamic pressure also shows a wide energy spectrum that follows the Kolmogorov energy cascade (Fig. 7a). The maximum frequency of dynamic pressure is (again) at 1.25 Hz (that is a period of c. 800 ms), showing that the largest eddies also carry the largest dynamic pressure. In the wall region, the frequency peak associated with the maxima in dynamic pressure occurs again at 1.75 Hz. However, the frequency of the largest eddies at 1.25 Hz is still clearly visible and associated with the second highest values of dynamic pressure.

Our turbulence analysis highlights an important element of hazard generation inside pyroclastic surges. Due to their turbulent nature, pyroclastic surges show very strong fluctuations and a wide spectrum of dynamic pressures. The probability density functions of the dynamic pressure reveal the important role played by the turbulent excursions (Fig. 7b). The distributions are strongly skewed towards the high-pressure values. This shows that turbulent fluctuations in dynamic pressure can exceed the mean pressure by a factor of three to five (the ratio of maximum to mean dynamic pressures take values of 3.88 and 4.75 for the wall and jet regions, respectively).

The evidence of a wide spectrum of dynamic pressure and, in particular, the self-generation and downstream perpetuation of repeated high-pressure pulses inside pyroclastic surges need consideration when forecasting volcanic hazard. In turbulent flows, the most energetic frequency $f$ of turbulent oscillations is expressed by the Strouhal number[49,50]

$$\text{Str} = fL/U \quad (7)$$

where $L$ and $U$ are the currents' characteristic length and velocity scales.

In cases where the characteristic length scale $L$ is clearly defined by the geometry, as in the classical problem of the vortex shedding of variably turbulent flow around a cylinder[51], the frequency of the most energetic coherent structures can be predicted by the Strouhal number approaching a critical limit of Str ~ 0.3 at high values of Re ≥ $10^5$.

In our experimental pyroclastic surges, the characteristic length scale is set by the flow dynamics. The time-averaged flow height $L_{\text{ave}}$ is c. 1.2 m (that is the height of the gravity current's head and body regions), the depth- and time-integrated average flow

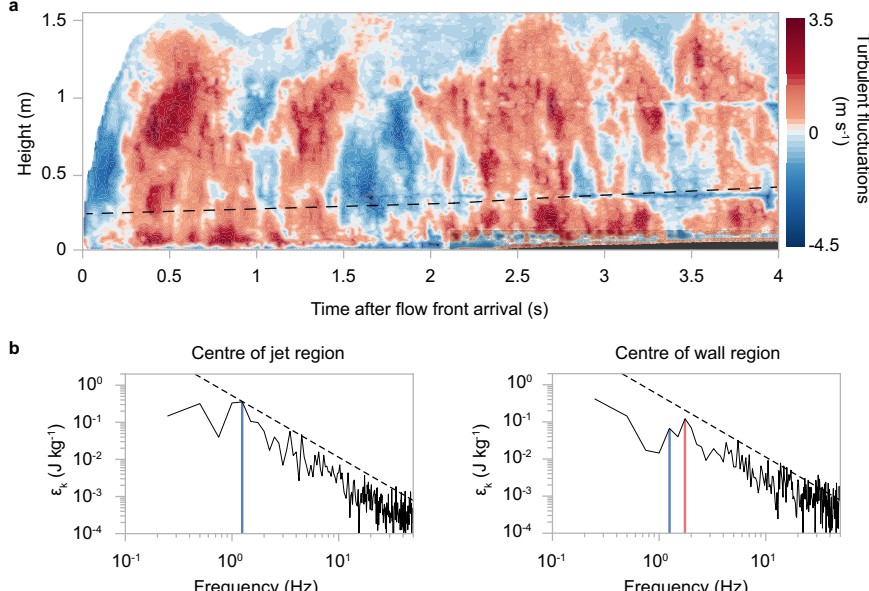

**Fig. 6 Coherent turbulence structure as the origin for flow oscillations. a** contour plot of turbulent fluctuations $u'$ (Eq. 3) against time at a static observer location of 3.12 m reveal long-period coherent turbulence structures of the experimental pyroclastic surge. The dashed line shows temporal variation of velocity maximum separating lower wall and upper jet regions. The aggrading deposit at the base of the flow is in dark grey. **b** spectra of specific kinetic energy $\varepsilon_k$ (Eq. 5) measured in the centre of the jet (left) and wall regions (right). The frequencies of the two energy peaks at 1.25 Hz (blue line) and 1.75 Hz (red line) are highlighted. The dashed black lines show the −5/3 Kolmogorov law of the inertial range of the turbulent energy cascade. The energy spectra are cut off at 50 Hz.

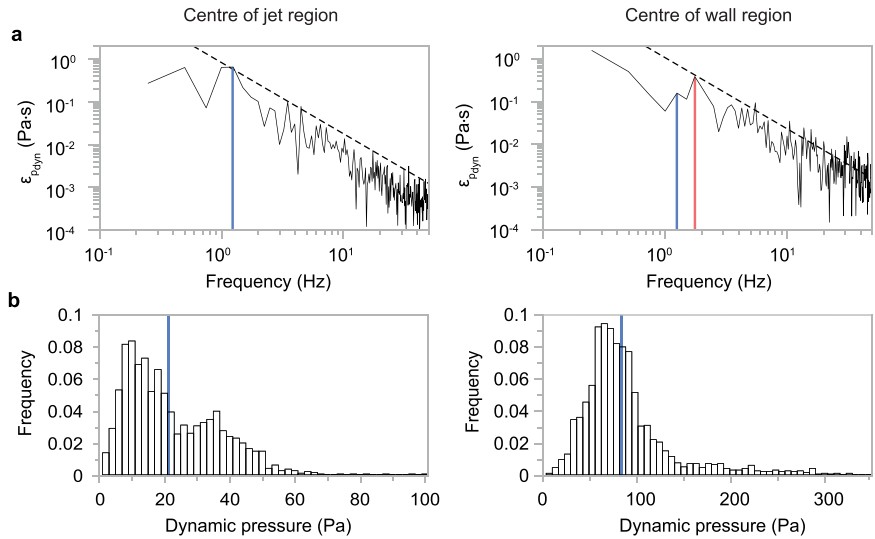

**Fig. 7 Turbulent excursions and focussing of dynamic pressure energy into large-scale coherent structures. a** energy spectra of dynamic pressure $\varepsilon_{P_{dyn}}$ (Eq. 6) in the centres of the jet (left) and wall regions (right). The dashed black lines show the -5/3 Kolmogorov law of the inertial range of the turbulent energy cascade. The energy spectra are cut off at 50 Hz. The frequencies of the two energy peaks at 1.25 Hz (blue line) and 1.75 Hz (red line) are highlighted. The 1.25 Hz peak, forming the energy maximum in the jet region and the second highest energy peak in the wall region, is associated with the largest coherent turbulence structures in the experimental pyroclastic surge. The 1.75 Hz peak, forming the energy maximum in the wall region, is related to internal gravity waves that propagate as pulses through the advancing current (Fig. 5). **b** histograms of probability density functions of dynamic pressure in the centres of the jet (left) and wall regions (right). Maximum values in dynamic pressure, which are associated with the largest coherent turbulence structures and internal gravity waves, exceed mean values of dynamic pressures, which are shown as blue vertical lines by a factor of around three to five.

velocity $U_{ave}$ is c. 4.8 m s⁻¹, and the frequency of the most energetic coherent structures associated with these characteristic lengths and velocity scales is $f = 1.25$ Hz as determined by Fourier analysis (Fig. 7a). Thus, at the experimental condition of Re ~ 1.5 × 10⁶, the Strouhal number takes a value close to the critical value at $Str$ ~ 0.31. This finding highlights that lower and upper flow boundary shear, which drive turbulent flow oscillation at the

largest flow length scale (that is $L_{ave}$), lead to the focussing of dynamic pressure energy inside the largest coherent turbulent structures that stretch across the flows' jet and wall regions (Fig. 6a).

However, for the assessment of pyroclastic surge hazards, it is important to also understand the physical process behind the largest dynamic pressures that are associated with the frequency

peak $f_{max} = 1.75$ Hz. The largest recurrent velocity scale in the experimental pyroclastic surges is the remarkably time-invariant velocity of the density discontinuities (pulses).

The two frequencies associated with the largest dynamic pressures, $f = 1.25$ Hz and $f_{max} = 1.75$ Hz, can be considered as the two characteristic velocity scales for propagation of the density current with characteristic length scale $L_{ave}$ over a distance of unit length. Let us call $c$ the characteristic velocity associated with $f_{max}$. Dimensional arguments (see Supplementary Note 1) suggest

$$f_{max}/f = c/U_{ave} \quad (8)$$

Solving for $c$ gives a velocity of 6.73 m s$^{-1}$, which corresponds with the experimentally measured average velocity for the propagating density discontinuities (6.72 m s$^{-1}$). This shows that the maximum dynamic pressure is carried by the propagating density discontinuities.

Density discontinuities in shallow flows travel at the velocity of gravity waves. Gravity waves were hypothesized to cause flow pulsing in PDCs[43]. For hazard considerations, the dimensional analysis can be taken slightly further (see Supplementary Note 1) to express the ratio of maximum to mean dynamic pressures in pyroclastic surges as

$$\frac{P_{max}}{P_{ave}} = \frac{\rho_{max}c^2}{\rho_{ave}U_{ave}^2} \sim \left(\frac{c}{U_{ave}}\right)^4 = 3.84 \quad (9)$$

This value corresponds closely with the experimentally measured ratio $P_{max}/P_{ave}$ in the wall region of 3.88.

This analysis suggests that two phenomena generate recurrent low-frequency oscillations in the experimental pyroclastic surges: the largest-scale, shear-generated coherent structures (the largest eddies) and gravity waves.

**Turbulence-enforced destructiveness in real-world flows.** The ferocity of real-world pyroclastic density currents usually prohibits estimating equivalent flow properties. However, the catastrophic pyroclastic surge-forming eruption of Whakaari (White Island) on 9 December 2019 was recorded by four cameras from different angles and distances and through the seismo-acoustic array of the national monitoring network GeoNet on the island. In the aftermath of the c. 82 s duration series of phreatic explosions, we were able to retrieve high-resolution (1-s interval) data of flow height and flow front velocity for the entire flow runout of c. 1700 m. Over the initial c. 930 m of runout, the velocity of the surges was remarkably constant with a time-average of $U_{ave}$ of c. 17 m s$^{-1}$, while several measurements of the characteristic flow height give an average value of $L_{ave}$ of c. 26 m.

At a radial distance from vent of c. 800 m, the south-southwesterly edge of the surges billowed over a ridge and overran one of the sensor stations equipped with an infrasound microphone (Fig. 8a). This resulted in the recording of the flow pressure in the flows' jet region that shows the occurrence of regular pressure pulses (Fig. 8a). The Fourier analysis of the pressure signal identifies an energy spectrum that approximately follows the Kolmogorov cascade up to around 30–40 Hz, while at higher frequencies, the limited frequency response of the infrasound sensor is dominated by white noise. However, at the high-energy start of the energy cascade of the inertial range, the most energetic frequency peak of the pressure spectrum clearly emerges from the cascade and has a value of $f = 0.199$ Hz (Fig. 8a). Combing this frequency with the measurements of $L_{ave}$ and $U_{ave}$ through Eq. 7 provides an estimate of the Strouhal number for the highly turbulent flow of Str ~ $0.30 \pm 0.02$.

We are not aware of any other direct measurements into pyroclastic density currents. However, numerical multiphase

simulations of the directed blast from Mount St Helens in 1980 provided an estimate of the flow pressure in the lower flow region[52]. In the case of the modelled flow runout along the east-northeast sector, the time-averaged flow height is approximately 750 metres. Flow front velocities were obtained by ref. [53] and have a mean value of $U_{ave}$ of c. 93 m s$^{-1}$. The modelled signal of dynamic pressure is characterized by the occurrence of four clear pressure pulses over a flow duration of c. 114 s (that is a frequency of c. 0.035 Hz; Fig. 20 in ref. [52]). These numerical model results suggest a characteristic Strouhal number of Str ~ $0.28 \pm 0.03$, which is close to the critical value. We note however that the combination of direct measurements and numerical modelling data is not without ambiguity and this estimate needs to be viewed with caution.

In order to analyse additional direct measurements into high-energy, highly turbulent gas-particle currents, we examined the measurements of the powder snow avalanche #20163017 recorded at the field experimental site Vallée de la Sionne, Switzerland[54]. The lateral edge of this dilute fully turbulent avalanche engulfed an instrumented pole structure, where at a height of c. 16 metres above the ground, a pitot tube recorded the flow pressure of the bypassing current (Fig. 8b)[55]. The time-averaged flow height during passage of this avalanche region $L_{ave}$ was c. 20 m. Estimates of the time-averaged flow velocity were obtained through signal time correlation of paired near-infrared sensors and yield $U_{ave}$ of c. 40 m s$^{-1}$. The Reynolds number ranges from $10^6$–$10^9$ [55]. As in the large-scale experiments and in the pyroclastic density currents from Whakaari, the pressure signal in the jet region of the avalanche is characterized by a series of regular pressure pulses (Fig. 8b). The Fourier analysis of the time series of pressure yields a frequency of the most energetic pressure fluctuations of $f = 0.6$ Hz (Fig. 8b). Together with measurements of $L_{ave}$ and $U_{ave}$, this gives a characteristic Strouhal number of, again, Str ~ 0.3.

The above results suggest that, in high-Reynolds number dilute pyroclastic density currents and in powder snow avalanches, the most energetic frequency of turbulent fluctuations is characterised by a critical Strouhal number of approximately 0.3. It is thus similar to the critical value of Str seen in the classic fluid mechanics experiments of ref. [51] and in recent work on turbulent plumes[56]. Systematic series of high-resolution numerical simulations scaled to the conditions of pyroclastic density currents can provide further complexity to this finding. However, these results lend themselves to propose a simple model to predict the frequency of high-energy turbulent oscillations and thus the number of pulses of high dynamic pressure (and pulses of high velocity and/or flow density) that characterise pyroclastic surges. Equation 7 can be rewritten to solve for the number of pressure pulses per minute as

$$N = 60 \text{Str} \frac{U}{h} \quad (10)$$

where Str ~ 0.3.

The local number of pulses in the jet regions of a number of dilute pyroclastic density currents, in the PELE experiments and snow avalanche #20163017 can thus be predicted through estimates of characteristic flow velocity and flow height and typical variability (Fig. 9). This demonstrates that the bypassing of pyroclastic surges over static observer locations (e.g. infrastructures, shelters etc.) is characterised by c. 2–20 low-frequency pressure pulses per minute due to oscillation of the largest eddy scale. Slower pyroclastic density currents tend to show higher numbers of dynamic pressure pulses than faster currents.

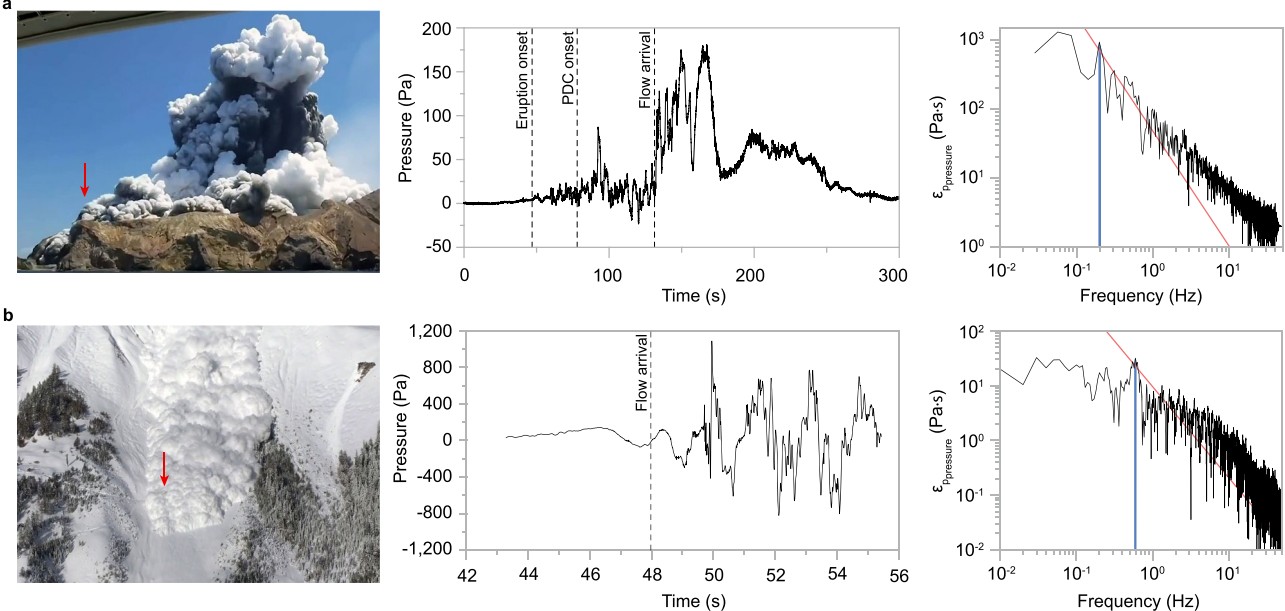

**Fig. 8 Direct measurements of pressure inside natural pyroclastic surges and snow avalanches.** The first pressure measurements inside pyroclastic density currents during the December 9 2019 eruption of Whakaari (White Island) in New Zealand in comparison with pressure measurements inside powder snow avalanche #20163017 in Vallée de la Sionne (Switzerland). **a** left: view to the West of Whakaari volcano during the December 9 2019 eruption with the collapsing eruption column in the centre-right of the image and the pyroclastic surges advancing from right to left along the crater floor and billowing over the crater walls. The red arrow marks the location of the WIZ infrasound sensor situated at the far ridge on the south-western side of the crater floor. Middle: time series of pressure recorded by an infrasound sensor. Vertical dashed lines indicate the times of eruption onset, onset of PDC propagation at the base of the collapsing eruption column and PDC arrival at the sensor. Right: energy spectrum of pressure $\varepsilon_{P_{dyn}}$. The blue vertical line highlights the most energetic frequency at 0.199 Hz. **b** left: frontal view of powder snow avalanche #2016301755. The red arrow marks the location of the pylon-mounted pitot tube sensor that becomes engulfed by the left edge of the avalanche. Middle: time series of pressure recorded by the pitot tube. The vertical dashed line indicates the times of avalanche arrival at the sensor. Right: energy spectrum of pressure $\varepsilon_{P_{dyn}}$. The blue vertical line highlights the most energetic frequency at 0.6 Hz. Red lines mark the -5/3 Kolmogorov law of the inertial range of the energy cascade. Part a image courtesy of Allessandro Kauffmann (top); part b image courtesy of Bettina Sovilla (bottom).

## Discussion

The first measurements of the turbulence structure of pyroclastic surges in large-scale experiments and during the 9 December 2019 Whakaari eruption add critical complexity to our understanding of how their hazard impacts are generated and perpetuated during flow runout. Our results demonstrate that during pyroclastic surge propagation, the dynamic pressure (that is the kinetic energy per unit volume) generated by the conversion of potential energy is distributed across a wide range of frequencies. This creates a spectrum of kinetic energies that is strongly skewed towards high dynamic pressures. Importantly, the maximum pressures exceed the mean values, which are routinely estimated for volcanic hazard assessments, by a factor of at least three. To prevent underestimation of hazard impacts, we strongly suggest that this factor is applied to traditional estimates of dynamic pressure using bulk current properties[13–16,57]. For example, while, for two- to three-story buildings, a mean dynamic pressure of 5–10 kPa leads to failure of only door and window building elements, the at least three-times larger maximum pressures of 15–30 kPa cause significantly higher damage and probable failure of exterior building walls made of brick, stone or concrete[58].

The skewed distribution of dynamic pressure arises from the largest coherent turbulence structures and internal gravity waves that generate the high-energy start of the turbulent spectrum. These occur as markedly periodic and low-frequency flow characteristics that define the engines and top of a turbulent energy cascade through which large turbulent fluctuations diffuse to the smallest scales.

The largest-scale coherent turbulent structures have characteristic length scales of the current height and their main driver

is shear with the lower and upper flow boundaries. Inside these structures, low-frequency oscillations of flow velocity, density and dynamic pressure are strongly correlated in time. With regards to hazards, this time correlation implies the compounding of hazard impacts from concurrent peaks in: high dynamic pressure causing destruction impacts to infrastructure; high density causing suffocation impacts as well as burn impacts (because heat is concentrated in the particle phase of the multiphase flows); and high velocity. The rapid succession of these compounded hazard impacts in the form of low-frequency oscillations is likely to exacerbate damage.

The internal gravity waves constitute the fastest recurrent flow structures. These propagate downstream through the pyroclastic surge as discontinuities; they carry the largest dynamic pressures and they are most dominant in the flow's wall region, that is the region where humans are likely to live and build infrastructure.

Our results show that the kinetic energy peaks associated with the largest-scale coherent structures are bounded at the lowest frequency by a critical Strouhal number around 0.3. This allows the prediction of the number of dynamic pressure pulses for natural flow scales (through Eq. 10; Fig. 9). However, the exact mechanism of formation of the internal gravity waves cannot be detected by our experimental method and needs further experimental and numerical investigation of pyroclastic surges with a wide range of density contrasts. Possible mechanisms include the formation of weak shocks and supersonic instabilities during column collapse e.g.[59,60] or the steepening and breaking of internal gravity waves that potentially form during the development of a strong vertical density stratification immediately after collapse[32].

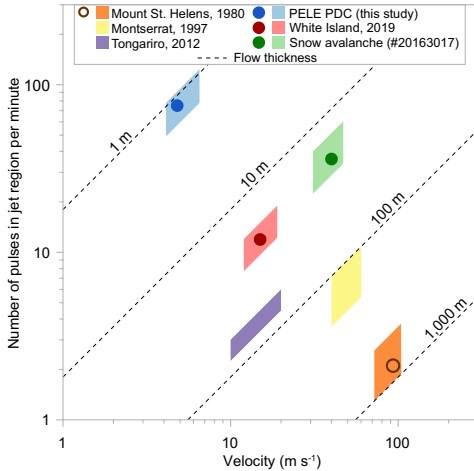

**Fig. 9 Occurrence and prediction of pressure pulses in real-world flows.**
The number of dynamic pressure pulses associated with the passage of the largest-scale coherent turbulence structures in natural pyroclastic density currents, large-scale experiments, and snow avalanche #20163017. Measured data for large-scale PELE experiments, Whakaari (White Island) surges and the snow avalanche are shown as closed circles. Measured data in numerical simulation of Mount St. Helens blast is shown as open circle. Modelled numbers of pressure pulses (Eq. 10) are shown as polygons based on published data of flow velocity and flow height from the 1980 blast of Mount St. Helens (United States)[52,66], the 1997 Boxing day blast on Montserrat (West Indies)[67], and the 2012 blast-like surges from Tongariro (New Zealand)[68] and snow avalanche #20163017 in Vallée de la Sionne (Switzerland)[55]. The dashed lines indicate flow thickness. Model input data are summarised in Supplementary Table 2.

Finally, the downstream propagation and perpetuation of the coherent turbulence structures and gravity waves demonstrate how dynamic pressure energy, and therefore hazard potential, is effectively transported in pyroclastic surges outwards from source. The largest-scale coherent turbulent structures and the internal gravity waves carry the bulk of the flow energy and flow detructiveness. This highlights that evaluation of hazard models needs to be informed by the turbulence structure of pyroclastic surges. These findings are important to hazard decision-making in volcanic regions globally. The turbulent and destructive processes described here are not only relevant for pyroclastic surges but for other particle-laden gravity currents as well, including powder snow avalanches, turbidity currents and dust storms.

## Methods

**Large-scale experiments**. The eruption simulator PELE (the Pyroclastic flow Large-scale Experiment, fully described in ref. [24]) is an international test facility where we can synthesize, view and measure inside the highly dangerous interior of pyroclastic density currents. Experimental currents of up to 6 tonnes of natural volcanic material and gas reach velocities of 7–32 m s$^{-1}$, flow thicknesses of 2–4.5 m, and runouts of >35 m[24]. PELE synthesizes experimental pyroclastic density currents by the controlled gravitational collapse of variably diluted suspensions of pyroclastic particles and gas from an elevated hopper onto an instrumented runout section. PELE is operated indoors, inside a 16 m high, 25 m long and 18 m wide disused boiler house. The apparatus contains four main structural components: (i) Tower. A 13 m high structure that lifts either a 4.2 m$^3$ hopper (for moderate to high discharge rates of 300 to 1500 kg s$^{-1}$) or a 0.7 m$^3$ hopper (for low discharge rates from 30 to 200 kg s$^{-1}$) to the desired discharge height. The hoppers include internal hopper heating units to bring the pyroclastic material to target temperatures of up to 400 °C, which is directly measured by thermocouples, and they are mounted on four load cells to capture the time-variant mass discharge. (ii) Column. A ≤ 9 m high shroud through which the air-particle mixtures accelerate under gravity. (iii) Chute. A 12 m long multi-instrumented channel section, variably adjustable to slope angles between 5 and 25° and with 0.6–1.8 m high sides of temperature-resistant glass. (iv) Outflow. A 25 m long flat instrumented runout section that extends outside the building. The physical characteristics of the gas-particle suspensions prior to impact (velocity, mass flux,

volume flux, particle concentration, temperature), the solids components (grain-size distribution, density), and boundary conditions (substrate roughness, slope, channel width) can be controlled to generate a wide range of reproducible natural conditions[24]. For the experiments reported in this study, we used the small hopper of 0.7 m$^3$ to generate a fully turbulent density current with a basal bedload region, but without a dense underflow, which would occur at intermediate to large discharge rates in the large hopper setup condition. The input and boundary conditions for the reported experiments are given in Supplementary Table 1.

The use of volcanic material and air in our experiments ensures natural stress coupling between the solid and fluid phases. The volcanic material, involving particle sizes from 2 μm to 16 mm, consists of a blend of two standardised ignimbrite deposits F1 and F2 from the 232 CE Taupo eruption[31]. The first component (F1) is a proximal medium-ash-dominated ignimbrite deposit with a unimodal grain-size distribution, a median diameter of 366 μm, and 4.5 wt.% of extremely fine ash (<63 μm). The second component (F2) is a fine ash rich facies from the base of the proximal Taupo ignimbrite, showing a polymodal distribution, median diameter of 103 μm, and 36.5 wt.% extremely fine ash. The experiments reported here used a material blend with F1=60 wt.% and F2=40 wt.% (see the grain-size distribution in Supplementary Fig. 2) yielding a mixture with 20 wt.% of particles smaller than 63 μm.

The resulting pyroclastic density current analogues are fully turbulent with Reynolds numbers up to 10$^6$ (and up to 10$^7$ in proximal regions). Dimensionless products quantifying the scaling similitude of natural and experimental currents for the bulk flow are shown in Table 1. Further details of the experimental protocol, properties of the volcanic material, and measurement techniques are described elsewhere[24], but some measurements and analytical methods specific to the results presented here are detailed below.

**Sensors and analytical methods**. Twenty fast cameras (60–120 frames per second), and three normal-speed cameras (24–30 frames per second) positioned at different distances, viewing angles and directions, recorded the downstream evolution of the experimental pyroclastic density current. At runout distances of 3.12 m, 5.77 m, and 10.9 m, three digital high-speed and high-resolution cameras (NAC Hotshot at a framerate of 500 frames per second) recorded the flow passage of the lower 1.2 to 1.8 m of the flow capturing the aggrading deposit, the bedload region, the entire body region and part of the wake region of the turbulent gravity current. The tempered glass walls of the channel were illuminated by arrays of LED floodlights, which allowed for a detailed analysis of the gas-particle transport and sedimentation processes with particle image velocimetry (PIV; using the algorithm PIVlab[61]). Two-dimensional velocity fields were obtained with PIV from the high-speed footage at 2 ms time intervals. At a runout distance of 17.8 m, we obtained vertical velocity profiles through PIV using the footage from a high-resolution thermal infrared camera operated at 100 Hz.

At the static observer locations at 3.12 m, 5.77 m and 10.9 m, vertical arrays of transparent sediment samplers collected the flowing mixture. During the experiment, we record with high-resolution high-speed cameras the sequential filling of the flow samplers. They are open on the upstream side allowing the flow to enter through the 1.69 cm$^2$ cross-sectional area while on the downstream side, a 16 microns mesh allows only the gas-phase of the flow to exit, leading to accumulation of the transported particles. From this we derive continuous data of flow sediment volume passing a position as a function of time. In addition, we use the PIV results from high-speed camera recordings giving the downslope flow velocity component at a position 5 cm upstream of each flow sampler. We measure the weight and density of the accumulated material for selected time intervals to calculate the time-variant porosity of the captured sediment, as well as the particle grain-size distributions. Particle solids-concentrations $C_s$ are defined as follows

$$C_s(z, t) = \frac{V_d(1 - \varepsilon)}{u A_o t} \quad (11)$$

where $V_d$ is the time-variant accumulated sediment volume inside the flow sampler, $u$ the time-variant velocity obtained by PIV at the entrance of the flow sampler, $A_o$ the cross-sectional area of the flow sampler, $t$ the selected time interval and $\varepsilon$ the time-variant sediment porosity.

In addition to the time ($t$)-variant and height ($z$)-variant flow velocity $u(z, t)$, grain-size distributions, volumetric particle concentrations $C_S(z, t)$, we obtain vertical profiles of time-variant and height-variant flow temperature $T(z, t)$ from vertical arrays of fast thermocouples. These time-series data allow for the calculation of dynamic pressure inside the flow, defined as

$$p_{dyn}(z, t) = \frac{1}{2}\rho_C|u|^2 \quad (12)$$

$$\rho_C = C_S\rho_S + \frac{p_a}{RT}(1 - C_S) \quad (13)$$

$$R = y_g R_g \quad (14)$$

where $z$ is height in the slope-perpendicular direction, $\rho_C(z, t)$ is the bulk density of the current, $\rho_S$ is the particle density, $p_a$ is the ambient pressure, $T$ is the temperature, $y_g$ is the mass fraction of the gas components (including moisture), and $R_g$ is their gas constant.

The mean velocity profile defined in Eq. (2) has four time-dependent parameters: the dimensionless height $\eta = z/h_m(t)$, with $h_m$ being the height of the wall region where the velocity maximum $U_m$ occurs; $\xi(t)$ the wall layer exponent; and $\chi(t)$ is the jet region exponent, where $U_m$, $h_m$, $\xi$ and $\chi$ are fitted variables.

The temporal dependence of the fit is smoothed by using polynomial functions for these parameters. We tested different combinations, up to the fifth polynomial degree. This shows that satisfactorily results can be obtained using a third-degree polynomial for $U_m(t)$, while a first-order polynomial is sufficient for the remaining three parameters.

For the computation of turbulence energy spectra, we downsample the time-series data to 250 Hz. This ensures that the data used for the Fourier analysis, which yield the spectra presented in Figs. 6 and 7, are above the Nyquist frequency. In our results, we relate frequency peaks and pattern of the Fourier spectra to their underlying physical mechanisms, such as frequencies of the most energetic coherent turbulence structures, the energy cascade of the inertial range or the frequency associated with the fast-travelling density discontinuities. However, the finite duration of the time series of velocity and pressure examined by Fast Fourier analysis also generates amplitude/frequency datapoints that are not related to physical processes. For instance, for the experimental data shown in Figs. 6b and 7a, the 0.5 Hz and the 0.25 Hz datapoints are the two lowest frequencies of the Fourier analysis. These are the frequencies associated with the periods of 2 and 4 s, respectively, which are the consequence of the selected time window of 4 s used in the Fourier analysis. The selection of a finite length of a time series for a Fourier analysis is akin to filtering the data with a threshold function. The two lowest frequency datapoints calculated in as Fourier transform have periods equal to and half of the length of the selected time window.

**Field measurements of pyroclastic surges at Whakaari (White Island, New Zealand).** The December 9, 2019 eruption of Whakaari was captured by four cameras capturing the eruption from different distances and angles with framerate of one image per second. Three-dimensional locations of eruption features were projected using a projection matrix that was solved through the method of ref. [62] and calibrated using known image (2D) to real-world (3D) point correspondences. This allowed the mapping of the positions and thicknesses of the propagating surge front as a function of time. Flow pressure was recorded by an infrasound microphone sensor located at the WIZ sensor station of the national monitoring agency GeoNet. The sensor is located on a ridge at c. 90 m above sea level at a distance of c. 800 m from the vent. The pressure time-series data has a temporal resolution of 10 ms. For the computation of the energy spectrum of pressure, we selected a 35 s time window starting from the arrival of the pyroclastic surge at the WIZ station at 01:13:11.313130 UTC. The data-series is downscaled to 50 Hz, ensuring the Fourier analysis, which yields the energy spectrum presented in Fig. 8a, is above the Nyquist frequency.

**Field measurements of powder snow avalanche #20163017 at Vallée de la Sionne, Switzerland.** The snow avalanche #20163017 was artificially triggered in 2016 at the Vallée de la Sionne in Switzerland[55]. The time series of air pressure in the jet region of the avalanche was recorded by a pitot tube sensor installed at 16 m above ground at a resolution of 5 kHz[54]. The effective height above ground for this event was 14 m, due to a 2 m thick snow deposit on the ground. For the computation of the energy spectrum of pressure, we selected a 7.46 s time window starting with the arrival of the avalanche at the sensor. The data series is downscaled to 2500 Hz, ensuring the Fourier analysis, which yields the energy spectrum presented in Fig. 8b, is above the Nyquist frequency.

## Data availability

The data generated in this study are available at https://doi.org/10.5281/zenodo.5635370. Pressure measurements of pyroclastic surges at Whakaari (White Island, New Zealand) are available from the New Zealand national monitoring agency GeoNet. Pressure measurements of powder snow avalanche #20163017 from Vallée de la Sionne (Switzerland) are available at https://doi.org/10.5281/zenodo.1415456.

## Code availability

The code used to produce the turbulence analysis is freely available at https://www.python.org.

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

## Acknowledgements

The authors thank A. Moebis and K. Kreutz for assistance during the experiments. E. Meiburg, K. Arentsen and G. Lube Sr are thanked for internal review. Allessandro Kauffmann is thanked for providing the image of the 2019 Whakaari eruption. This study was supported by the Royal Society of New Zealand Marsden Fund (contract no. MAU1902), the New Zealand Ministry of Business, Innovation and Employment's Endeavour Fund (contract no. RTVU1704) and Resilience to Nature's Challenges Science Challenge Fund (GNS-RNC047). E.C.P.B and J.D. contributions were supported by the National Science Foundation (grant no. EAR 1852569).

## Author contributions

E.B. and G.L. designed the experiments and wrote the first draft of the manuscript, which was then revised by all the authors. E.B. and G.L. conducted and analysed the experiments with the help of M.C. and L.F. and they interpreted the data together with M.C., T.E.-O., E.C.P.B., J.D. and B.S. G.L developed the PELE facility.

## Competing interests

The authors declare no competing interests.
