## [Peer Review File · Nature Communications]

REVIEWER COMMENTS

Reviewer #1 (Remarks to the Author):

Review of:

Destructiveness of pyroclastic surges controlled by turbulent fluctuations

Authors: Brosch, Lube, Cerminara, Esposti-Ongaro, Breard, Dufek, Sovilla, and Fullard

Reviewer: Benjamin J. Andrews, Smithsonian Global Volcanism Program

Summary

Pyroclastic surges, the dilute end-member of pyroclastic density currents, constitute a substantial volcanic hazard. These currents can destroy nearly anything in their paths, including reinforced buildings. Here the authors use large-scale experiments and field measurements to quantify the dynamic pressures within surges. Importantly, they show that turbulent structures within the currents drive dynamic pressure maxima downstream as a series of low-frequency structures. The magnitude of pressure within these currents can be many times greater than the mean dynamic pressure. The authors show that the frequency of these large turbulent structures is set by a Strouhal number of ~ 0.3 – this means that the dynamics of the largest structures (and thus the pressure waves) can be predicted from observations of natural currents. The paper very nicely shows that fluctuations in dynamic pressure manifest through two different processes, shear-induced production of large eddies and gravity waves, resulting in two different (and predictable) frequencies and magnitudes of pressure fluctuation.

This paper makes important contributions to our understanding of pyroclastic density current dynamics and the hazards that these currents can pose. Quantifying the range in dynamic pressure that PDCs can generate is useful for academic studies of PDCs, but has real-world implications for hazard modelling and mitigation. The paper does a good job of linking flow descriptors (such as the turbulent energy spectrum) that are perhaps esoteric to most readers to tangible flow properties (such as dynamic pressure). Along those lines, the methods are sound and well-described. Finally, the presentation of natural PDC data from Whakaari is excellent: quantifying the eruption characteristics is very important and useful, providing a unique dataset.

Areas for Improvement

There are a few areas where the paper could be improved. Towards the end of the discussion the authors describe the length scale of the eddies as being roughly equal to the flow depth – although this makes sense for the vertical length scale, it does not work for the streamwise length. Instead, the authors could present this as the ratio of the mean velocity to the period of pressure oscillations shows that the largest eddy length scales are ~ 4 times greater in the streamwise direction than the flow depth (this is discussed in more detail in the comment for line 394).

In summary, I recommend acceptance with minor revisions.

Benjamin Andrews, Smithsonian Global Volcanism Program

Line-by-line comments

General note – suggest changing “flow-internal” to “flow” throughout the manuscript

30 – add “also called dilute pyroclastic DENSITY currents”

30-31 – suggest rephrasing as “is a critical and outstanding uncertainty in volcanic...”

31-32 – rephrase as “Over 100 million people worldwide are potentially endangered by these fast moving (10s to 100s of m/s), fully turbulent and ground-hugging...”

35 – suggest “and forests results from internal dynamic pressures of 10s to 100s of kPa and remains poorly mitigated globally.”

37 – is “perpetuate” the right word here? That suggests that the pressures are remaining elevated (and even \sim constant?) over great distances. How about “how large pressures manifest or persist over...” (apologies for wordsmithing...)

39 – change “Nevertheless, the...” to “But the violence...” – this makes it clear that we have not been able to measure these quantities precisely because PDCs are so violent.

41 – change to “our knowledge of the highly destructive dynamic pressures inside flows...”

73 – delete the comma after “both”

75 – I suggest changing this to “we refine our understanding of...” – to me, saying that you add

complexity makes the problem worse, whereas you have shown that although the real world is complex, it follows certain patterns and can thus be forecast.

82 – change “damage-causing” to “high”

115 – phrase as “ambient air entrainment”

117 – Can you clarify that the “density contrast” here means the ratio of current to air density, rather than the difference.

132-133 – why should the rough boundary of the flume floor affect or mitigate the boundary effects of the glass walls? (the smooth wall makes sense to minimize the boundary effects)

155 – I suggest rephrasing as “... for pyroclastic surges have large uncertainties...” (this is a little more neutral phrasing than “plagued”)

169 – I think that the comma after “exponent” should be a semicolon

192-193 – this is an interesting (and important) point about the fluctuating dynamic pressures. Although it is surprising, it probably should not be. By this I mean that previous experimental studies have described sedimentation waves and persistent fluctuations in particle concentration (both of which record the turbulent transport of particles and non-uniform particle concentration) – but those studies didn’t make the connection between velocity and concentration (density) to demonstrate that there should be big fluctuations in dynamics pressure. (so good work that the authors have made this connection)

244 – rephrase this sentence as “The largest eddies, generated by shear between the top of the current and atmosphere, form the high-energy start of the spectra and energy cascade.”

251 – change “destructiveness” to “destruction”

265 – delete “violent” from “violent turbulent excursions”

267 – rephrase as “in dynamic pressure can exceed the mean pressure by a factor of three to five...”

290 – “for the assessment of pyroclastic surge hazards...”

309 – very nice set of work here

315 – delete “with certainty”

318 – rephrase as “aftermath of the ~82 second duration series of phreatic...”

325 – change to “overran”

332 – should this be “ $f=0.199$ Hz”? that is, does it need units?

374 – change “infrastructures, shelters, etc.” to “infrastructure, buildings, etc.”

376 – please clarify if the higher number of pulses in slow currents is a higher total number or a higher frequency (N/time) – I think that you probably mean the latter

381 – I suggest changing to “... eruption improve our understanding of how...” (this is from the same reasoning as at line 75)

383 – “during pyroclastic surge propagation”

390 – I would rephrase this sentence. It isn’t so much that energy is focused into the largest eddies, but rather that those large structures form the energy generating part of the turbulent spectrum.

394 – the discussion or description of the turbulent length scale here sounds like the integral turbulent length scale. The paper has presented very nice and sophisticated analyses of the frequency of the oscillations. But I am curious if you were to apply a simple treatment of the mean velocity and integral length scale to obtain the integral timescale ($U/L=\tau$) if you would obtain a similar timescale or frequency for the oscillations. As a rough calculation, it looks like $\sim 6\text{m/s} / 1.2\text{m} = 200$ ms, which is a bit faster than the 800 ms period for the observed pressure oscillations. This treatment is admittedly very simple, but it seems to suggest that although the eddies have a vertical lengthscale of ~ 1.2 m (the flow depth), they are elongate in the streamwise direction. An alternative way to look at this would be to say that the ratio of the flow mean velocity and characteristic time scale (~ 800 ms) shows that the characteristic length scale of the eddies is not the flow depth, but rather something on the order of 5 m.

405 – change to “flows’ wall regions” (plural on regions)

416 – change to “...waves carry the bulk of the flow energy...”

417 – change to “evaluation” (singular) and “needs” (s on the end)

419 – “The turbulent and destructive processes described here are not only relevant for pyroclastic surges but for other particle laden gravity currents as well, including powder snow...”

422 – as a minor quibble, tornadoes are certainly particle-laden and turbulent, but they are not density currents

462 – change to “up to 10^6 (and up to 10^7 in proximal regions).”

613 – can delete the superscript “and” from the three references

627 – change “perpetuate” to “persist”

682 – for figure 7b, change the y-axis labels to “frequency” or “occurrence” or something similar. Labeling them as density makes me think that these are showing a relationship between current

density (kg/m³) and dynamic pressure.

Reviewer #2 (Remarks to the Author):

I think the manuscript of Brosch et al. is an interesting piece of work dealing with large scale experiments of pyroclastic density currents. In my opinion, the main outcome of the work is the experimental description of density pulses within diluted PDCs. They were hypothesised on the basis of sedimentological and stratigraphic arguments (Sulpizio et al., 2007; Sulpizio and Dellino, 2008), but never described in lab experiments. This is a valuable results that sheds light into the physics of these complex multiphase flows.

For this reason, I think to focus the attention of the reader to the destructiveness of these flows distracts from the main outcome. Indeed, destructiveness with respect to what? Buildings? Infrastructure? environment? Human health? The discussion about the impact of these flows is very general in the manuscript, and I would suggest to condense all the considerations about hazard in an ancillary paragraph.

Other points

i) The internal dynamics of natural PDCs were already recorded and published in few papers (see for example Capra et al., 2018; Delle Donne et al., 2014). So, the data from natural flows here presented are not the first ones. In any case, the data from Colima and Montserrat may be converted into dynamic pressure and used for comparison in the manuscript.

ii) The vertical profile of velocity and concentration in diluted PDCs was already presented in Dellino et al., (2008; 2010) and Doronzo and Dellino (2011). I would suggest to compare the presented experimental profile with that derived from sedimentological models and validated using large scale experiments (Dellino et al., 2010)

iii) I would appreciate some more details in the way Eq. 2 was derived

iv) Statement in lines 190-193 is not true. Pulses within PDCs are part of current sedimentological models (see Sulpizio et al, 2007; 2010; Sulpizio and Dellino 2008).

v) Dynamic pressure is the combination of velocity and density, so it is not strange it shows the same pulsating behaviour that the two components (velocity and density)

vi) gravity waves were already considered as responsible of formation of pulses within PDCs by Sulpizio and Dellino (2008)

Minor points

- Line 175-176: please, explain better what you mean saying "However, we note that instantaneous and time-integrated profiles of particle concentration obey self-similar forms"

-Line 384: kinetic energy is not equivalent of dynamic pressure

Reference cited

- Capra et al. 2018. Bulletin of Volcanology, 80:34 <https://doi.org/10.1007/s00445-018-1206-4>
Delle Donne et al. 2014. Geological Society, London, Memoirs, 39, 167–176.
<http://dx.doi.org/10.1144/M39.9>
Dellino et al. 2008. J. Geophys. Res. 113, B07206. doi:10.1029/2007JB005365
Dellino et al. 2010. Earth and Planetary Science Letters 295, 314–320
Doronzo and Dellino 2011. Earth and Planetary Science Letters 310, 286–292
Sulpizio et al., 2007. Sedimentology 54, 607–635
Sulpizio et al., 2010. Bull. Volcanol. doi:10.1007/s00445-009-0340-4.
Sulpizio and Dellino 2008. In: Gottsman, J., Marti, J. (Eds.), Calderas Volcanism: Analysis, Modeling and Response. Elsevier, Amsterdam

Sincerely
Roberto Sulpizio

Reviewer #3 (Remarks to the Author):

Reviewer: Olivier Roche

The authors present a study on turbulence fluctuations in pyroclastic surges. They combine the results of large-scale experiments with measurements in natural flows, a pyroclastic surge and a powder snow avalanche. The data on the pyroclastic surge, acquired during the 9 December 2019 Whakaari eruption, are the first of their kind. The main conclusion of the study is that the dynamic pressure energy is carried by large-scale turbulent structures and gravity waves occurring at a specific frequency, and pressure maxima are several times equal to the mean values. This has important implications for hazard assessment.

This is an interesting paper, which I enjoyed reading. It presents a coherent set of results that may be appealing to different scientific communities working on turbulent geophysical flows. Please find below comments whose objective is to help clarify some sections of the text and which, I hope, will help to improve the manuscript.

Large-scale experiments

- Heating the volcanic material in the hopper over a period of three days was probably enough for thermal equilibration, as stated by the authors, but was the temperature measured inside the hopper to make sure that the volcanic material was actually at a temperature of 120°C before running the experiment?
- A temperature of 120°C is lower than that of most natural pyroclastic surges. I think it would be interesting to discuss briefly the possible consequences of higher temperatures in term of decrease of flow density through air entrainment and the possible differences with the results of the large-scale experiment.
- Is the grain size distribution of the volcanic material actually bimodal? Supplementary Fig. 1a does show a maximum at grain size $\phi=6.5$ but it is very moderate.

Scaling

I am not sure there is an "excellent match" of the values of the dimensionless numbers (lines 142-149). Since the experiments involve a natural material but they are ran at small length-scale and velocity compared to nature, then there are inherent scaling issues. I agree that the values of some dimensionless numbers do match (Richardson, thermal Richardson, Froude, Rouse) but the ranges of values of the Reynolds, Stokes and Stability numbers in the experiments are in the lower limit of those of natural flows. This is probably not a severe limitation of the study, but a short discussion on this issue would allow the reader to better appreciate the significance of the experiments.

Pulsing in pyroclastic surges

- Is pulsing only self-generated in experiments? I agree that pulses can emerge in a surge generated under downstream steady conditions, but I wonder if some unsteadiness may not be generated when the volcanic material released from the hopper impacts the channel (regardless the discharge rate). We can draw a parallel here with the simulations of eruptive column collapse with which the authors are familiar: steady eruptive conditions can lead to unsteady flow dynamics due to the complex process of accumulation of the volcanic material on the ground, which generates successive pulses. Can the authors document the impact of the granular material in the channel? Could pulses control, at least partially, the frequency of the observed velocity, density, and pressure oscillations? I think this is an important point and a few sentences about this would be very helpful.
- The time interval of the regular oscillations of c. 800 ms is not obvious for the flow velocity in Fig. 3 (though it is clear in Fig. 4), should the authors use another color scale? The data of flow density in Fig. 3 suggest that the time interval between the pulses decreases with time down to about 500 ms.
- The authors describe regular density discontinuities (or pulses) with an average period of c. 750 ms (line 215). This is a very interesting observation; does this correspond to the regular density oscillations of c. 800 ms shown in Fig. 3?
- The data in Figs. 6 and 7 suggest another energy peak at about 0.5 Hz for the jet region. Why don't you consider this frequency? In lines 280-293, why do you consider $f=1.25$ Hz in the jet

region and not $f=1.75$ Hz also shown for the wall region to discuss the Strouhal number, while you consider only $f=1.75$ Hz to discuss the largest dynamic pressures?

- I am confused with the energy spectra shown in Figs. 7 and 8. In both cases you show ϵ_{Pdyn} , which, if I understand well, is calculated from Eq. 6 for the experiments (Fig. 7) or is obtained from Fourier analysis of the pressure signal in natural surges (Fig. 8). Is it the same parameter ϵ_{Pdyn} ?

Surges in nature and powder snow avalanche

- It would be fair to state in the main text the research organization that acquired the data of the pyroclastic surges at Whakaari. Is it the New Zealand national monitoring agency GeoNet as mentioned in the Data Availability section?

- In Fig. 8, the measured pressure is almost entirely positive in the volcanic surge while both negative and positive pressures are reported for the snow avalanche. Is this due to the different nature of the sensors used (i.e. infrasound sensor in the surge and a Pitot tube in the avalanche) or possibly to different flow dynamics?

Methods

I am confused with Eq. 13 used to define the bulk density of the current, which is used to calculate the dynamic pressure with Eqs. 1 and 12. What is the pressure p in Eq. 13? Is the bulk flow density equal to c. 3.4 kg/m^3 as mentioned in line 116?

This study has important implications for assessment of natural hazards. In addition to the interesting data shown in Fig. 9 for the number of pulses per minute, could the authors comment briefly on the possible maximum dynamic pressures in natural surges, taking into account ranges of natural flow densities and velocities? Are these maximum pressures likely to be about three times the mean values, as shown in experiments, or is the pressure ratio likely to be different, and if so, why? How would these maximum pressures compare with the typical resistance of buildings?

L23. Say rather "hazard assessment" (as in line 28)?

L30. "also called dilute pyroclastic currents"?

L56. State that the term polydispersity means large particle size distribution.

L106. Do these 4-8 mm rock pebbles confer the substrate roughness of 5 mm given in Supplementary Table 1?

L116. Say rather "density ratio"?

L131-133. How do combined rough bottom surface, smooth side-walls and fast flow velocities minimize sidewalls effects? Please justify.

L139-140. Please define u as well.

L187-188. Please see general comments on time interval of oscillations.

L206-207. A unimodal source mass discharge rate does not imply absence of oscillations. Please see general comments.

L208-207. This is a very interesting observation.

L235-237. This sentence is not clear. Does it mean that shear generates the largest eddies?

L262-269. This is another important information.

L280-293. Please see general comment.

L294. In Fig. 7 the maximum dynamic pressure is shown for the jet region at $f=1.25$ Hz. Please clarify.

L301-302. This is a good point.

L335. I think you are right.

L343-345. Agreed. It is a wise statement.

L351-353. Can you give an estimate for the Reynolds number of the powder snow avalanche?

L392. Typo (characteristics).

L474. Is "high-temperature" glass walls the correct wording?

L484-485. What is the mesh size? Can fine particles pass through the mesh?

L501. $\rho_c(z,t)$

L607. It could be worth stating that this is the solid mass fraction measured at a given time.

Fig 2b. Show the particle concentration or flow density (cf. Fig. 3) instead of the solid mass fraction?

Supplementary Table 2. Please give also the Reynolds numbers, if possible.

We would like to thank all reviewers for their supportive reviews. Please find our replies and changes made to the reviewers' comments outlined below.

Reviewer 1: Benjamin Andrews

Pyroclastic surges, the dilute end-member of pyroclastic density currents, constitute a substantial volcanic hazard. These currents can destroy nearly anything in their paths, including reinforced buildings. Here the authors use large-scale experiments and field measurements to quantify the dynamic pressures within surges. Importantly, they show that turbulent structures within the currents drive dynamic pressure maxima downstream as a series of low-frequency structures. The magnitude of pressure within these currents can be many times greater than the mean dynamic pressure. The authors show that the frequency of these large turbulent structures is set by a Strouhal number of ~ 0.3 – this means that the dynamics of the largest structures (and thus the pressure waves) can be predicted from observations of natural currents. The paper very nicely shows that fluctuations in dynamic pressure manifest through two different processes, shear-induced production of large eddies and gravity waves, resulting in two different (and predictable) frequencies and magnitudes of pressure fluctuation. This paper makes important contributions to our understanding of pyroclastic density current dynamics and the hazards that these currents can pose. Quantifying the range in dynamic pressure that PDCs can generate is useful for academic studies of PDCs, but has real-world implications for hazard modelling and mitigation. The paper does a good job of linking flow descriptors (such as the turbulent energy spectrum) that are perhaps esoteric to most readers to tangible flow properties (such as dynamic pressure). Along those lines, the methods are sound and well-described. Finally, the presentation of natural PDC data from Whakaari is excellent: quantifying the eruption characteristics is very important and useful, providing a unique dataset.

We thank the reviewer for their positive review and constructive comments.

Areas for Improvement

There are a few areas where the paper could be improved. Towards the end of the discussion the authors describe the length scale of the eddies as being roughly equal to the flow depth – although this makes sense for the vertical length scale, it does not work for the streamwise length. Instead, the authors could present this as the ratio of the mean velocity to the period of pressure oscillations shows that the largest eddy length scales are ~ 4 times greater in the streamwise direction than the flow depth (this is discussed in more detail in the comment for line 394).

Please see our reply in the specific line comment (line 394) below.

Line-by-line comments

General note – suggest changing “flow-internal” to “flow” throughout the manuscript

Reply. We agree.

Changes. As suggested, we exchanged ‘flow-internal’ to ‘flow’ throughout the manuscript.

30 – add “also called dilute pyroclastic DENSITY currents”

Reply. We agree.

Changes. We changed ‘pyroclastic current(s)’ to ‘pyroclastic density current(s)’ throughout the manuscript.

30-31 – suggest rephrasing as “is a critical and outstanding uncertainty in volcanic...”

Reply. We agree.

Changes. We rephrased this in lines (now) 31.

31-32 – rephrase as “Over 100 million people worldwide are potentially endangered by these fast moving (10s to 100s of m/s), fully turbulent and ground-hugging...”

Reply. We agree.

Changes. We rephrased this sentence accordingly in lines (now) 32.

35 – suggest “and forests results from internal dynamic pressures of 10s to 100s of kPa and remains poorly mitigated globally.”

Reply. We agree.

Changes. We iterated the sentence in question as suggested in lines (now) 36.

37 – is “perpetuate” the right word here? That suggests that the pressures are remaining elevated (and even ~constant?) over great distances. How about “how large pressures manifest or persist over...” (apologies for wordsmithing...)

Reply. That’s a good suggestion.

Changes. We changed the wording in lines (now) 38 to now read “...large dynamic pressures manifest or persist over long flow runouts...”

39 – change “Nevertheless, the...” to “But the violence...” – this makes it clear that we have not been able to measure these quantities precisely because PDCs are so violent.

Reply. That’s a good suggestion.

Changes. We iterated the sentence as suggested in lines (now) 40.

41 – change to “our knowledge of the highly destructive dynamic pressures inside flows...”

Reply. We agree with the suggesting re-wording.

Changes. We iterated the sentence as suggested in lines (now) 43.

73 – delete the comma after “both”

Reply. Done.

Changes. We deleted the comma after ‘both’ in line (now) 78.

75 – I suggest changing this to “we refine our understanding of..” – to me, saying that you add complexity makes the problem worse, whereas you have shown that although the real world is complex, it follows certain patterns and can thus be forecast.

Reply. That’s a good suggestion.

Changes. We reworded the sentence as suggested, which in lines (now) 80 reads: “Through this, we refine our understanding of hazard impacts...”.

82 – change “damage-causing” to “high”

Reply. That’s a good suggestion.

Changes. We changed the wording as suggested in lines (now) 87.

115 – phrase as “ambient air entrainment”

Reply. That’s a good suggestion.

Changes. We rephrased the above to as suggested in lines (now) 122.

117 – Can you clarify that the “density contrast” here means the ratio of current to air density, rather than the difference.

Reply. We agree with the suggested word change.

Changes. We changed ‘density contrast’ to ‘density ratio of the flow with the ambient’ in lines (now) 124.

132-133 – why should the rough boundary of the flume floor affect or mitigate the boundary effects of the glass walls? (the smooth wall makes sense to minimize the boundary effects)

Reply. The reviewer is correct, the bottom roughness has no direct effect here.

Changes. We rewrote this sentence to read in line (now) 139: “In our experiments, we minimize these effects through the use of hydraulically smooth side-walls (laminar layer thickness/wall roughness <5) and through the flow’s high Reynolds number ($Re=1.5\times 10^6$), which is inversely related to the thickness of the viscous boundary layer.”

155 – I suggest rephrasing as “... for pyroclastic surges have large uncertainties...” (this is a little more neutral phrasing than “plagued”)

Reply. We agree.

Changes. We iterated the sentence as suggested in lines (now) 165.

169 – I think that the comma after “exponent” should be a semicolon

Reply. We agree.

Changes. We changed the comma to a semicolon in line (now) 179.

192-193 – this is an interesting (and important) point about the fluctuating dynamic pressures. Although it is surprising, it probably should not be. By this I mean that previous experimental studies have described sedimentation waves and persistent fluctuations in particle concentration (both of which record the turbulent transport of particles and non-uniform particle concentration) – but those studies didn’t make the connection between velocity and concentration (density) to demonstrate that there should be big fluctuations in dynamics pressure. (so good work that the authors have made this connection)

Reply. Thank you.

Changes. No changes made.

244 – rephrase this sentence as “The largest eddies, generated by shear between the top of the current and atmosphere, form the high-energy start of the spectra and energy cascade.”

Reply. We agree.

Changes. We rephrased the sentence as suggested in line (now) 260.

251 – change “destructiveness” to “destruction”

Reply. We agree.

Changes. We changed the word as suggested in line (now) 268.

265 – delete “violent” from “violent turbulent excursions”

Reply. We agree

Changes. We deleted the word “violent” as suggested in line (now) 282.

267 – rephrase as “in dynamic pressure can exceed the mean pressure by a factor of three to five...”

Reply. We agree.

Changes. We rephrased the sentence as suggested in line (now) 284.

290 – “for the assessment of pyroclastic surge hazards...”

Reply. We agree.

Changes. We reworded the sentence as suggested in line (now) 308.

315 – delete “with certainty”

Reply. We agree.

Changes. We deleted “with certainty” as suggested in line (now) 337.

318 – rephrase as “aftermath of the ~82 second duration series of phreatic...”

Reply. We agree.

Changes. We rephrased the sentence as suggested in line (now) 340.

325 – change to “overran”

Reply. We agree.

Changes. We changed the word as suggested in line (now) 347.

332 – should this be “ $f=0.199$ Hz”? that is, does it need units?

Reply. Yes, indeed, the units are Hz.

Changes. We added the missing unit ‘Hz’ in line (now) 354.

374 – change “infrastructures, shelters, etc.” to “infrastructure, buildings, etc.”

Reply. We agree.

Changes. We exchanged the ‘shelters’ with ‘infrastructure’ as suggested in line (now) 398.

376 – please clarify if the higher number of pulses in slow currents is a higher total number or a higher frequency (N/time) – I think that you probably mean the latter

Reply. Correct, the correct term here is ‘frequency’.

Changes. We clarified this in line (now) 400, which now reads: “Slower pyroclastic density currents tend to show higher frequencies of dynamic pressure pulses than faster currents.”

381 – I suggest changing to “... eruption improve our understanding of how...” (this is from the same reasoning as at line 75)

Reply. We agree.

Changes. We clarified the sentence as suggested in line (now) 405.

383 – “during pyroclastic surge propagation”

Reply. We agree.

Changes. We changed the above as suggested in line (now) 407.

390 – I would rephrase this sentence. It isn't so much that energy is focused into the largest eddies, but rather that those large structures form the energy generating part of the turbulent spectrum.

Reply. We agree with the suggestion.

Changes. The rephrased sentence in line (now) 419 now reads: “The skewed distribution of dynamic pressure arises from the largest coherent turbulence structures and internal gravity waves that generate the high-energy start of the turbulent spectrum.”

394 – the discussion or description of the turbulent length scale here sounds like the integral turbulent length scale. The paper has presented very nice and sophisticated analyses of the frequency of the oscillations. But I am curious if you were to apply a simple treatment of the mean velocity and integral length scale to obtain the integral timescale ($U/L=\tau$) if you would obtain a similar timescale or frequency for the oscillations. As a rough calculation, it looks like $\sim 6\text{m/s} / 1.2\text{m} = 200\text{ ms}$, which is a bit faster than the 800 ms period for the observed pressure oscillations. This treatment is admittedly very simple, but it seems to suggest that although the eddies have a vertical lengthscale of $\sim 1.2\text{ m}$ (the flow depth), they are elongate in the streamwise direction. An alternative way to look at this would be to say that the ratio of the flow mean velocity and characteristic time scale ($\sim 800\text{ ms}$) shows that the characteristic length scale of the eddies is not the flow depth, but rather something on the order of 5 m.

Reply. The streamwise lengths of the most energetic coherent turbulence structures, D , can be calculated directly from the time-series presented in Fig. 4 of the velocity in the jet region. At the static observer location of 3.12 m, D has an average value of 3.2 m, confirming that the coherent structures are elongated in the downstream direction. Furthermore, D systematically decreases during flow runout and at a distance of 18 m, D takes an average value of 1.6 m. We agree with the reviewer, that these findings of the shape of the coherent turbulence structures are worthwhile reporting in this work.

However, D is not a characteristic length-scale in the determination of the typical frequency (and associated energies) of the most energetic coherent turbulence structures. Indeed, we measure this frequency directly. Using the above determined value of D at 3.12 m in Eq. 7 yields a frequency of c. 0.43 Hz. Fig. 6 and 7 show, that there is no energy peak associated with this frequency, and that this frequency is also not associated with the turbulent energy cascade. The energy peaks closest to 0.43 Hz are 0.25 Hz and 0.5 Hz and correspond to the duration of the dataset (4 seconds) used in the Fourier analysis.

Please note that the proposed simple scaling ($U/L=\tau$) implies a Str number of unity which we show is not the case for the highly turbulent flows in question.

Changes. In line (now) 210, we added: "This is associated with a downstream decay of the streamwise length-scale of the oscillations from c. 3.2 to 1.6 m, while the integral vertical scale (that is the height of the flow body) remains relatively constant with an average thickness of 1.2 m."

405 – change to "flows' wall regions" (plural on regions)

Reply. We agree.

Changes. We corrected the above in line (now) 435.

416 – change to "...waves carry the bulk of the flow energy..."

Reply. We agree.

Changes. We iterated the sentence as suggested in line (now) 449.

417 – change to "evaluation" (singular) and "needs" (s on the end)

Reply. We agree.

Changes. We corrected the above as suggested in line (now) 450.

419 – "The turbulent and destructive processes described here are not only relevant for pyroclastic surges but for other particle laden gravity currents as well, including powder snow..."

Reply. We agree.

Changes. We rephrased the sentence as suggested in line (now) 452.

422 – as a minor quibble, tornadoes are certainly particle-laden and turbulent, but they are not density currents

Reply. We agree.

Changes. We thank for pointing this out and excluded 'tornadoes' in line (now) 457.

462 – change to "up to 10^6 (and up to 10^7 in proximal regions)."

Reply. We agree.

Changes. We changed the above as suggested in line (now) 498.

613 – can delete the superscript "and" from the three references

Reply. We agree.

Changes. We deleted the 'and' from the references in line (now) 677.

627 – change “perpetuate” to “persist”

Reply. We agree.

Changes. We changed the word to 'persist' as suggested in line (now) 691.

682 – for figure 7b, change the y-axis labels to “frequency” or “occurrence” or something similar. Labeling them as density makes me think that these are showing a relationship between current density (kg/m³) and dynamic pressure.

Reply. We agree

Changes. In figure 7b, we changed the label of the y-axis from 'Density' to 'Frequency'.

Reviewer 2: Roberto Sulpizio

I think the manuscript of Brosch et al. is an interesting piece of work dealing with large scale experiments of pyroclastic density currents. In my opinion, the main outcome of the work is the experimental description of density pulses within diluted PDCs. They were hypothesised on the basis of sedimentological and stratigraphic arguments (Sulpizio et al., 2007; Sulpizio and Dellino, 2008), but never described in lab experiments. This is a valuable results that sheds light into the physics of these complex multiphase flows. For this reason, I think to focus the attention of the reader to the destructiveness of these flows distracts from the main outcome. Indeed, destructiveness with respect to what? Buildings? Infrastructure? environment? Human health? The discussion about the impact of these flows is very general in the manuscript, and I would suggest to condense all the considerations about hazard in an ancillary paragraph.

Reply. We thank the reviewer for the thoughtful comments and valuable review.

As stated by reviewers 1 and 3, (this work) “has real-world implications for hazard modelling and mitigation” and “has important implications for hazard assessments”. We certainly agree with these statements and therefore think that it is important to keep the discussion about the hazard implications or our results.

In the discussion, the hazard implications for dynamic pressure impacts to infrastructure is discussed in lines (now) 411-418 (to which we now added an example of the difference in destructiveness at mean and maximum values of dynamic pressure to buildings, as suggested by reviewer 3), while the compounding of suffocation, burn and dynamic pressure hazards due to the strong time-correlation of the low-frequency velocity and density oscillations are discussed in lines (now) 425-432.

Changes. Lines (now) 411-418 now read “Importantly, the maximum pressures exceed the mean values, which are routinely estimated for volcanic hazard assessments, by a factor of at least three. To prevent underestimation of hazard impacts, we strongly suggest that this factor is applied to traditional estimates of dynamic pressure using bulk current properties^{13, 14, 15, 16, 57}. For example, while,

for two- to three-story buildings, a mean dynamic pressure of 5–10 kPa leads to failure of only door and window building elements, the at least three-times larger maximum pressures of 15–30 kPa cause significantly higher damage and probable failure of exterior building walls made of brick, stone or concrete⁵⁸.”.

Lines 427-432 read “With regards to hazards, this time-correlation implies the compounding of hazard impacts from concurrent peaks in: high dynamic pressure causing destruction impacts to infrastructure; high density causing suffocation impacts as well as burn impacts (because heat is concentrated in the particle phase of the multiphase flows); and high velocity. The rapid succession of these compounded hazard impacts in the form of low-frequency oscillations is likely to exacerbate damage.”

Other points

i) The internal dynamics of natural PDCs were already recorded and published in few papers (see for example Capra et al., 2018; Delle Donne et al., 2014). So, the data from natural flows here presented are not the first ones. In any case, the data from Colima and Montserrat may be converted into dynamic pressure and used for comparison in the manuscript.

Reply. In lines 68 and 337, we state that the pressure measurements inside the 2019 Whakaari PDCs constitute the first direct measurements inside dilute PDCs. We acknowledge the studies of seismic and acoustic signals **induced by** PDCs (in the ground and atmosphere surrounding the flows). However, these signals do not represent direct internal measurements inside the flows, but rather need to be qualitatively interpreted based on conceptual models of the PDC structure to infer aspects of the flow dynamics. We are not aware of any concept or model that allows for the ‘conversion’ of seismo-acoustic signals in the ground and atmosphere surrounding PDCs into signals of flow internal pressure inside PDCs.

Changes. In line (now) 74, we added the following statement: “These direct measurements inside PDCs are distinct from previous geophysical signals induced by PDCs in the ground and atmosphere surrounding the flow (e.g. Capra et al., 2018; Delle Donne et al., 2014; Scharff et al., 2019). Such seismic, acoustic and radar signals currently require interpretation based on models of the PDC structure and interaction with the environment, to infer aspects of the flow dynamics.

ii) The vertical profile of velocity and concentration in diluted PDCs was already presented in Dellino et al., (2008; 2010) and Doronzo and Dellino (2011). I would suggest to compare the presented experimental profile with that derived from sedimentological models and validated using large scale experiments (Dellino et al., 2010)

Reply: As suggested by the reviewer, we compared the model presented by Dellino et al. 2008 using PYFLOW2.0 implemented by Dioguardi et al. (2017) against our experimental data. In PYFLOW2.0, we used the following input parameters for the experimental data, which are derived from our experimental measurements:

Parameter	Value	
MU	1.9E-5	Viscosity of the fluid (Pa s)
DENGAS	1.12	Fluid density (kg m ⁻³)
MODEL	TWOLAYERS	

PROBT	0.05	T-test confidence
ZLAM	0.01	Thickness of layer (m)
C0	0.55	Solid concentration in layer
KS	0.006	Roughness dimension (m)
DENS_ENT	700	Solid density (kg m ⁻³)
DM_ENT	0.005	Clast diameter (m)
RHOS (1,0)	2385	Solid density (kg m ⁻³)
D50MM(1)	0.08	Mean grain-size (mm)
SORTING(1)	1.6	Sorting of grain size (phi)
NCLASS(1)	1	
CDLAW(1)	DIOGMELE	Type of drag law
SHAPEFACT(1,0)	0.5	Mean shape factor

Comparison of modelled vertical profiles of (a) particle concentration, (b) velocity and (c) dynamic pressure against experimental measurements (black lines). The green, blue, and red lines are three different model results based on the 16th, 50th and 84th percentile of the flow grain-size distribution, respectively.

The modelled vertical profiles of flow particle concentration, velocity and dynamic pressure deviate significantly from the experimental measurements.

First, the modelled velocity profile is not realistic beyond the wall region. This occurs because the PYFLOW model extends the Law of the Wall across the entire flow height instead of depicting a Gaussian decay in the jet region.

Second, the concentration is underpredicted by at least three orders of magnitude. This poor fit occurs because the model assumes that particle settling velocities are independent of concentration and that individual particle motion is unaffected by the motion of the fluid and other particles in its surrounding. This assumption of a homogenous system is applied to dilute water-particle suspensions with typical density ratios of c. 1–2, where inertial and viscous drag are dominant (Middleton and Southward, 1984).

However, fluid-particle transport in PDCs is dominated by kinematic and gravitational decoupling and non-homogeneous dynamics, owing to their high gas-particle density ratio (500–1000), high fine-ash content and low viscosity of the dusty gas phase. This leads to the formation of preferential concentration (one-way coupling) and mesoscale particle clusters (two and four-way coupling) which create a heterogeneous system (Brosch and Lube, 2020). The main effect of the self-organization of particles in clusters is the reduction of the effective drag coefficient on particles (Wang and Maxey, 1994; Aliseda et al. 2002; Capecelatro et al. 2014). In dilute gas-particle systems this leads to an increased settling velocity of particles by a factor of up to 2.5 (Breard et al., 2016, Lube et al. 2020, Li et al. 2021).

In addition, the PYFLOW model relies on the approach developed by Rouse for water-particles flows (e.g. Middleton and Southward, 1984), where due to the mass loading of c. 1–2 compared to 500–2000 in gas-particles flows (e.g. PDCs), preferential concentration and clustering will be much less important.

Third, due to the poor fit of the velocity and concentration profiles, the form of the modelled dynamic pressure profile is unrealistic and significantly underpredicts measured values of dynamic pressure for most of the flow depth.

A test of the models for vertical profiles of velocity and concentration in diluted PDCs of Dellino et al. (2008; 2010) and Doronzo and Dellino (2011) is beyond the scope of this article. However, we suggest that this test is pursued by the volcanological community, because of the frequent use of these models for hazard assessments.

Changes. No changes made.

iii) I would appreciate some more details in the way Eq. 2 was derived

Reply. We agree.

Changes. In line (now) 179 we added: “Equation (2) is a differentiable version of the profile proposed by Cantero-Chinchilla et al. (2015), who proposed a power law in the boundary layer and a Gaussian profile in the outer layer.

iv) Statement in lines 190-193 is not true. Pulses within PDCs are part of current sedimentological models (see Sulpizio et al, 2007; 2010; Sulpizio and Dellino 2008).

Reply. We agree that the concept of pulsing as envisaged in sedimentological models could be mentioned here.

Changes. In line (now) 205, we added the following sentence: “However, the occurrence of flow-internal pulses has been recorded in sedimentological studies (e.g. Sulpizio et al, 2007; 2010; Sulpizio and Dellino 2008).

v) Dynamic pressure is the combination of velocity and density, so it is not strange it shows the same pulsating behaviour that the two components (velocity and density)

Reply. We agree and, in line (now) 197, we state: “Due to the temporal correlation of the velocity and density oscillations, the time-variant dynamic pressure is also characterized by the passage of marked pressure oscillations that show the same regular period of c. 800 milliseconds.”

Changes. No changes made.

vi) gravity waves were already considered as responsible of formation of pulses within PDCs by Sulpizio and Dellino (2008)

Reply. Thank you for alerting us to this point. We agree that this reference is worthwhile citing here. However, please note that the work of Sulpizio and Dellino (2008) considers pulsing due to gravity waves formation in dense granular flow regimes, rather than in the fully dilute, fully turbulent transport system considered here.

Changes. In line (now) 322, we added to the first sentence to now read: “Density discontinuities in shallow flows travel at the velocity of gravity waves. Gravity waves were hypothesized to cause flow pulsing in PDCs (Sulpizio and Dellino, 2008).”

Minor points

- Line 175-176: please, explain better what you mean saying "However, we note that instantaneous and time-integrated profiles of particle concentration obey self-similar forms"

Reply. We noticed that this sentence is not essential to explain the vertical velocity and density structure.

Changes. We deleted the sentence from lines (now) 187.

-Line 384: kinetic energy is not equivalent of dynamic pressure

Reply. We thank for pointing this out.

Changes. We clarified in lines (now) 407 the statement by correcting the relation between kinetic energy and dynamic pressure as follows: “Our results demonstrate that during pyroclastic surge propagation, the dynamic pressure (that is the kinetic energy per unit volume) generated by the conversion of potential energy is distributed across a wide range of frequencies.”

Reviewer 3: Olivier Roche

The authors present a study on turbulence fluctuations in pyroclastic surges. They combine the results of large-scale experiments with measurements in natural flows, a pyroclastic surge and a powder snow avalanche. The data on the pyroclastic surge, acquired during the 9 December 2019 Whakaari eruption, are the first of their kind. The main conclusion of the study is that the dynamic pressure energy is carried by large-scale turbulent structures and gravity waves occurring at a specific frequency, and pressure maxima are several times equal to the mean values. This has important implications for hazard assessment.

This is an interesting paper, which I enjoyed reading. It presents a coherent set of results that may be appealing to different scientific communities working on turbulent geophysical flows. Please find below comments whose objective is to help clarify some sections of the text and which, I hope, will help to improve the manuscript.

We thank the reviewer for valuable and constructive comments and suggestions.

Large-scale experiments

Heating the volcanic material in the hopper over a period of three days was probably enough for thermal equilibration, as stated by the authors, but was the temperature measured inside the hopper to make sure that the volcanic material was actually at a temperature of 120°C before running the experiment?

Reply. Yes, during heating, the temperature of the volcanic material is directly measured with a set of thermocouples inside the hopper to monitor thermal equilibration during the heating procedure.

Changes. In line (now) 473, we added "...to target temperatures of up to 400° C, which is directly measured by thermocouples, ..."

A temperature of 120°C is lower than that of most natural pyroclastic surges. I think it would be interesting to discuss briefly the possible consequences of higher temperatures in term of decrease of flow density through air entrainment and the possible differences with the results of the large-scale experiment.

Reply. In this study, we assure correct thermal scaling through the buoyant thermal energy density and the thermal Richardson number. The overlap of the values of both of these non-dimensional products in our large-scale experiments and in real-world flows, as shown in Table 1, assures their scaled similitude. The experimental temperature of 120°C was carefully chosen to be dynamically scaled correctly and to result in both buoyant and forced convection; to allow evaporation of moisture inside the pre-dried mixture over a period of three days (line (now) 96); and to constitute a suitable scaled intermediate temperature for the wide spectrum of temperatures in natural pyroclastic surges. A systematic test of the variation of temperature is, however, largely beyond the scope of this study, but an interesting topic for future investigations.

Changes. No changes made.

Is the grain size distribution of the volcanic material actually bimodal? Supplementary Fig. 1a does show a maximum at grain size $\phi=6.5$ but it is very moderate.

Reply. The grainsize distribution, with a main mode at 2 ϕ (250 μm) and a minor mode at 6.5 ϕ (11 μm), is weakly bimodal.

Changes. In line (now) 101, we expanded the description of the mixture to now read: "it has a weakly bimodal grain-size distribution ranging from 2 μm to 16 mm with a main mode at 250 μm and a minor second mode at 11 μm ."

Scaling

I am not sure there is an “excellent match” of the values of the dimensionless numbers (lines 142-149). Since the experiments involve a natural material but they are ran at small length-scale and velocity compared to nature, then there are inherent scaling issues. I agree that the values of some dimensionless numbers do match (Richardson, thermal Richardson, Froude, Rouse) but the ranges of values of the Reynolds, Stokes and Stability numbers in the experiments are in the lower limit of those of natural flows. This is probably not a severe limitation of the study, but a short discussion on this issue would allow the reader to better appreciate the significance of the experiments.

Reply. We would like to point out that, currently, the PELE setup is the only experimental facility globally where such an overlap and match in scaled similitude of experimental and real-world flows can be achieved. Expanding the ranges in Reynolds, Stokes and Stability numbers to their upper natural limits would mean to run large-scale experiments at orders of magnitudes larger length scales, using the full range of natural particle sizes and achieving order of magnitudes higher flow energies (to maintain flow and suspension of particles at close to natural scales). However, while not only unfeasible, this is not needed to achieve dynamic and kinematic scaling similarity. With regards to the Reynolds number it is important to reach full turbulence ($Re > 10^3$). In this work, we also demonstrate that Reynolds numbers above 10^5 are important to synthesize turbulent conditions of a limiting Strouhal number. Similarly, for the Stokes and Stability numbers it is important that all five regimes of gas-particle coupling and particle transport in eddies are realized, as it is the case in our experiments.

Changes. In line (now) 154, we exchanged “excellent” with “good” scaling. In line (now) 159, we added “The experimental ranges in Reynolds, Stokes and Stability numbers, together, ensure that the complete range of gas-particle feedback mechanisms and turbulent particle transport in eddies is realized.”

Pulsing in pyroclastic surges

Is pulsing only self-generated in experiments? I agree that pulses can emerge in a surge generated under downstream steady conditions, but I wonder if some unsteadiness may not be generated when the volcanic material released from the hopper impacts the channel (regardless the discharge rate). We can draw a parallel here with the simulations of eruptive column collapse with which the authors are familiar: steady eruptive conditions can lead to unsteady flow dynamics due to the complex process of accumulation of the volcanic material on the ground, which generates successive pulses. Can the authors document the impact of the granular material in the channel? Could pulses control, at least partially, the frequency of the observed velocity, density, and pressure oscillations? I think this is an important point and a few sentences about this would be very helpful.

Reply. In the simulation of eruptive column collapse, pulses in PDC generation derives from a recirculation pattern between the vertical volcanic jet and the collapsing current, which is not observed in the experiments. Other mechanisms are however possible. In the discussion section in line (now) 440, we stated: “However, the exact mechanism of formation of the internal gravity waves cannot be detected by our experimental method and needs further experimental and numerical investigation of pyroclastic surges with a wide range of density contrasts.” To guide future studies, we added two possibilities for potential generation mechanisms.

Change. In line (now) 442, we added: “Possible mechanisms include the formation of weak shocks and supersonic instabilities during column collapse (e.g. Sweeney and Valentine, 2017; Valentine and Sweeney, 2018) or the steepening and breaking of internal gravity waves that potentially form during the development of a strong vertical density stratification immediately after collapse (Brosch and Lube, 2020).

The time interval of the regular oscillations of c. 800 ms is not obvious for the flow velocity in Fig. 3 (though it is clear in Fig. 4), should the authors use another color scale? The data of flow density in Fig. 3 suggest that the time interval between the pulses decreases with time down to about 500 ms.

Reply. We agree with the reviewer that, due the time- and height-variance of the clearly oscillating velocity, density and dynamic pressure structure of the experimental pyroclastic surges, the approximate average oscillation period is slightly less discernable in contour plots than for instance in the time-series plots of Figure 4. We acknowledge that, depending on the color scale of the contour plots, the eye could be drawn to higher frequency pattern in certain height regions of the flow. For this reason, in “Self-generated pulsing in pyroclastic surges”, we refer to the approximate oscillation period, while we quantify the dominant period of 800 milliseconds (1.25 Hz) in the velocity, dynamic pressure and hence density data explicitly through Fourier analysis. For comparison reasons, we choose the same contour color scale in all three plots of Figure 3. While changing color scales can enhance the visibility of the oscillation pattern, it tends to reduce data detail of the overall internal flow structure, which we want to illustrate in Figure 3.

Changes. No changes made.

The authors describe regular density discontinuities (or pulses) with an average period of c. 750 ms (line 215). This is a very interesting observation; does this correspond to the regular density oscillations of c. 800 ms shown in Fig. 3?

Reply. No, the density discontinuities do not correspond to the density oscillations. In line (now) 224, we clearly state that the density discontinuities occur in addition to the time-correlated velocity, density and dynamic pressure oscillations. Furthermore, in lines (now) 313-321, we show through dimensional arguments and experimental measurements that the oscillations and density discontinuities relate to two distinct characteristic velocity scales.

Changes. No changes made.

The data in Figs. 6 and 7 suggest another energy peak at about 0.5 Hz for the jet region. Why don't you consider this frequency? In lines 280-293, why do you consider $f=1.25$ Hz in the jet region and not $f=1.75$ Hz also shown for the wall region to discuss the Strouhal number, while you consider only $f=1.75$ Hz to discuss the largest dynamic pressures?

Reply. No, the 0.5 Hz data points seen in Figures 6 and 7 should not be considered for the Strouhal analysis of the oscillations of the most energetic turbulence structures. The 0.5 Hz and the 0.25 Hz data points are the two lowest frequencies in the Fourier analysis. These are the frequencies associated with the periods of two and four seconds respectively, which are the consequence of the selected time-window used in the Fourier analysis. The selection of a finite length of a time-series for a Fourier analysis is akin to filtering the data with a threshold function. The two lowest-frequency sinusoidal functions calculated in as Fourier transform have periods equal to and half of the length of the selected time-window. These two frequencies have thus nothing to do with turbulent energy cascade and the frequency of the most energetic coherent turbulent structures at the top of the cascade analyzed in this work.

The Strouhal number estimate is done only on the jet region, in analogy with considerations about the large-eddy frequencies in free shear turbulence reported at lines (now) 317-330. Near the wall, where the length scale is not set by the geometry but instead by the boundary layer dynamics, we prefer to use scaling considerations based on characteristic gravity wave velocity.

Changes. To make this methodological aspect of Fast Fourier Analysis more readily comprehensible to a wide audience, we added in line (now) 554 the following explanation to the Methods section:

“In our results, we relate frequency peaks and pattern of the Fourier spectra to their underlying physical mechanisms, such as frequencies of the most energetic coherent turbulence structures, the energy cascade of the inertial range or the frequency associated with the fast-travelling density discontinuities. However, the finite duration of the time-series of velocity and pressure examined by Fast Fourier analysis also generates amplitude/frequency datapoints that are not related to physical processes. For instance, for the experimental data shown in Figures 6b and 7a, the 0.5 Hz and the 0.25 Hz data points are the two lowest frequencies of the Fourier analysis. These are the frequencies associated with the periods of two and four seconds respectively, which are the consequence of the selected time-window of four seconds used in the Fourier analysis. The selection of a finite length of a time-series for a Fourier analysis is akin to filtering the data with a threshold function. The two lowest-frequency data points calculated in as Fourier transform have periods equal to and half of the length of the selected time-window.”

I am confused with the energy spectra shown in Figs. 7 and 8. In both cases you show ϵ_{Pdyn} , which, if I understand well, is calculated from Eq. 6 for the experiments (Fig. 7) or is obtained from Fourier analysis of the pressure signal in natural surges (Fig. 8). Is it the same parameter ϵ_{Pdyn} ?

Reply: The reviewer is correct that in both, Figure 7 and 8, the energy spectra shown are calculated as the Fourier transforms of the pressure data. In Figure 7, this is the Fourier transform of dynamic pressure (Eq.6), while in the cases of the natural flows these are the Fourier transforms of air-pressure as measured with the infrasound and pitot-tube sensors (see Methods).

Changes. We corrected the label of the y-axis in Fig. 8 from ϵ_{Pdyn} to $\epsilon_{Ppressure}$ and corrected the typo of the units of the amplitude axis to (Pa s).

Surges in nature and powder snow avalanche

It would be fair to state in the main text the research organization that acquired the data of the pyroclastic surges at Whakaari. Is it the New Zealand national monitoring agency GeoNet as mentioned in the Data Availability section?

Reply. This is correct. Thank you for pointing this out.

Changes. In addition to the statement on data availability, we clarified the data source in line (now) 339 as “... through the seismo-acoustic array of the national monitoring network GeoNet on the island.”

In Fig. 8, the measured pressure is almost entirely positive in the volcanic surge while both negative and positive pressures are reported for the snow avalanche. Is this due to the different nature of the sensors used (i.e. infrasound sensor in the surge and a Pitot tube in the avalanche) or possibly to different flow dynamics?

Reply. This is a good observation. The different signs of the infrasound pressure and pitot tube pressure measurements at Whakaari and in the Vallee de la Sionne are indeed related to the differences in the measurement principle and position within the flow. However, for the Fourier analysis (FFT), this difference is irrelevant. This is because the amplitude-frequency data of the FFT spectra are the absolute values of the complex number output of the FFT. Phase information of the signal are not needed to identify the frequency of the largest coherent turbulence structures.

Changes. No changes made.

Methods

I am confused with Eq. 13 used to define the bulk density of the current, which is used to calculate the dynamic pressure with Eqs. 1 and 12. What is the pressure p in Eq. 13? Is the bulk flow density equal to c. 3.4 kg/m^3 as mentioned in line 116?

Reply. Thank you for alerting us to this typo. We have corrected Eq. 13 and specified that p_a is the ambient pressure, which is $1.013 \times 10^5 \text{ Pa}$ for the experiment reported. As stated in lines (now) 121-124, the bulk flow density of c. 3.4 kg m^{-3} refers to the depth- and time-integrated flow density at a runout distance of 3.12 m.

Changes. In line (now) 541, we clarified "... p is the ambient pressure...".

This study has important implications for assessment of natural hazards. In addition to the interesting data shown in Fig. 9 for the number of pulses per minute, could the authors comment briefly on the possible maximum dynamic pressures in natural surges, taking into account ranges of natural flow densities and velocities? Are these maximum pressures likely to be about three times the mean values, as shown in experiments, or is the pressure ratio likely to be different, and if so, why? How would these maximum pressures compare with the typical resistance of buildings?

Reply. We agree with the reviewer on the critical implications of our findings for future hazard assessments as discussed in the final section. From our experimental measurements, which corresponds with the results of our dimensional analysis of the ratio maximum to mean dynamic pressure, we conclude, in the discussion in lines (now) 411-413, that a factor of at least three need to be applied to routinely estimated mean values of dynamic pressure using bulk current properties regardless of flow scale. It is also a good idea to provide an example of the difference in destructiveness at mean and maximum values of dynamic pressure to buildings. In line (now) 440, we state: "However, the exact mechanism of formation of the internal gravity waves cannot be detected by our experimental method and needs further experimental and numerical investigation of pyroclastic surges with a wide range of density contrasts."

Changes. In line (now) 415, we added the following statement: "For example, while, for two- to three-story buildings, a mean dynamic pressure of 5–10 kPa leads to failure of only door and window building elements, the at least three-times larger maximum pressures of 15–30 kPa cause significantly higher damage and probable failure of exterior building walls made of brick, stone or concrete (Valentine, 1998).

Line-by-line comments

L23. Say rather "hazard assessment" (as in line 28)?

Changes. We changed this in line (now) 23 to "hazard assessment".

L30. "also called dilute pyroclastic currents"?

Changes. We changed in line (now) 30 this to "(also called dilute pyroclastic currents)".

L56. State that the term polydispersity means large particle size distribution.

Reply. As suggested by the reviewer, we briefly explained the term polydispersity.

Changes. Line (now) 57 reads as “...polydispersity (large particle size distribution of the mixture)..”

L106. Do these 4-8 mm rock pebbles confer the substrate roughness of 5 mm given in Supplementary Table 1?

Reply. Thank you for pointing this out. Yes, the effective roughness equals to 5 mm. We have iterated the sentence.

Changes. Line (now) 112-113 read as “... Sub-rounded rock pebbles (4-8 mm in diameter) were glued to the channel base, generating an effective substrate roughness of 5 mm. This simulates, for the case of pyroclastic density currents with thicknesses of 50–500 m, a scaled c. 0.1–1 m-rough non-erodible volcanic surface.

L116. Say rather “density ratio”?

Changes. In Line (now) 124, we changed “density contrast” to “density ratio”.

L131-133. How do combined rough bottom surface, smooth side-walls and fast flow velocities minimize sidewalls effects? Please justify.

Reply. Please see our reply to the same point made by reviewer 1 in lines (now) 139-142.

Changes. We rewrote this sentence to read in line (139) as follows: “In our experiments, we minimize these effects through the use of hydraulically smooth side-walls (laminar layer thickness/wall roughness <5) and through the flow’s high Reynolds number ($Re=1.5\times 10^6$), which is inversely related to the thickness of the viscous boundary layer.”

L139-140. Please define $|u|$ as well.

Changes. We added the definition of $|u|$ in Line (now) 151: “... and $|u|$ (z, t) the magnitude of the local flow velocity.

L187-188. Please see general comments on time interval of oscillations.

Reply. Please see our clarifications to this comment further up in this document.

L206-207. A unimodal source mass discharge rate does not imply absence of oscillations. Please see general comments.

Reply. Please see our clarifications to this comment further up in this document.

L235-237. This sentence is not clear. Does it mean that shear generates the largest eddies?

Reply. Yes, the sentence states that boundary shear is generating the largest coherent turbulence structures.

Changes. In line (now) 252, we added a missing comma to clarify the sentence.

L280-293. Please see general comment.

Reply. Please see our clarifications to this comment further up in this document.

L294. In Fig. 7 the maximum dynamic pressure is shown for the jet region at $f=1.25$ Hz. Please clarify.

Reply. We don't understand this comment. In our view, the information provided in the text and Figure 7 is consistent.

L351-353. Can you give an estimate for the Reynolds number of the powder snow avalanche?

Reply. We added the Reynolds number information.

Changes. In line (now) 376, we added: "... of c. 40 ms^{-1} . The Reynolds number ranges from 10^6 - 10^9 ."

L392. Typo (characteristics).

Changes. We have corrected this typo in line (now) 422.

L474. Is "high-temperature" glass walls the correct wording?

Changes. In line (now) 510, we re-worded 'high-temperature' to 'tempered' glass.

L484-485. What is the mesh size? Can fine particles pass through the mesh?

Reply. The mesh size is 16 microns. While there is a small proportion of volcanic particles smaller than 16 microns, the air exiting the sampler is virtually particle free due to the strong reduction in air speed inside the sampler.

Changes: In line (now) 521, we added "...a 16 microns mesh...".

L501. $\rho_c(z,t)$

Changes. We have corrected this in line (now) 540 as indicated by the reviewer to $\rho_C(z, t)$

L607. It could be worth stating that this is the solid mass fraction measured at a given time.

Changes. In line (now) 671 we have included “at a given time” in the caption.

Fig 2b. Show the particle concentration or flow density (cf. Fig. 3) instead of the solid mass fraction?

Reply. The reason for us to present particle concentration as solid mass fraction is that it allows comparison of the experimental data with the empirical model of Cantero-Chinchilla et al. (2015) for turbidity currents.

Changes. No changes made.

Supplementary Table 2. Please give also the Reynolds numbers, if possible.

Reply. As suggested by the reviewer we have added bulk flow Reynolds numbers for the experimental and natural cases.

Changes. We updated Supplementary Table 2 by including a Reynolds number column.

References

- Aliseda, A., Cartellier, A., Hainaux, F. and Lasheras, J.C., 2002. Effect of preferential concentration on the settling velocity of heavy particles in homogeneous isotropic turbulence. *Journal of Fluid Mechanics*, 468: 77-105.
- Breard ECP, et al. Coupling of turbulent and non-turbulent flow regimes within pyroclastic density currents. *Nature Geoscience* 9, 767-771 (2016).
- Brosch, E. and Lube, G., 2020. Spatiotemporal sediment transport and deposition processes in experimental dilute pyroclastic density currents. *Journal of Volcanology and Geothermal Research*, 401: 106946.
- Cantero-Chinchilla, F.N., Dey, S., Castro-Orgaz, O. and Ali, S.Z., 2015. Hydrodynamic analysis of fully developed turbidity currents over plane beds based on self-preserving velocity and concentration distributions. *Journal of Geophysical Research-Earth Surface*, 120(10): 2176-2199.
- Capecelatro, J., Pepiot, P. and Desjardins, O., 2014. Numerical characterization and modeling of particle clustering in wall-bounded vertical risers. *Chemical Engineering Journal*, 245: 295-310.
- Capra, L., Sulpizio, R., Márquez-Ramirez, V.H., Coviello, V., Doronzo, D.M., Arambula-Mendoza, R. and Cruz, S., 2018. The anatomy of a pyroclastic density current: the 10 July 2015 event at Volcán de Colima (Mexico). *Bulletin of Volcanology*, 80(4): 34.
- Delle Donne, D., Ripepe, M., De Angelis, S., Cole, P.D., Lacanna, G., Poggi, P., Stewart, R., Wadge, G., Robertson, R.E.A. and Voight, B., 2014. Thermal, acoustic and seismic signals from pyroclastic density currents and Vulcanian explosions at Soufrière Hills Volcano, Montserrat, The Eruption of Soufrière Hills Volcano, Montserrat from 2000 to 2010. Geological Society of London, pp. 0.

- Dellino, P., Buttner, R., Dioguardi, F., Doronzo, D.M., La Volpe, L., Mele, D., Sonder, I., Sulpizio, R. and Zimanowski, B., 2010. Experimental evidence links volcanic particle characteristics to pyroclastic flow hazard. *Earth and Planetary Science Letters*, 295(1-2): 314-320.
- Dellino, P., Mele, D., Sulpizio, R., La Volpe, L. and Braia, G., 2008. A method for the calculation of the impact parameters of dilute pyroclastic density currents based on deposit particle characteristics. *Journal of Geophysical Research-Solid Earth*, 113(B7): B07206.
- Dioguardi, F. and Mele, D., 2018. PYFLOW_2.0: a computer program for calculating flow properties and impact parameters of past dilute pyroclastic density currents based on field data. *Bulletin of Volcanology*, 80(3): 28.
- Doronzo, D.M. and Dellino, P., 2011. Interaction between pyroclastic density currents and buildings: Numerical simulation and first experiments. *Earth and Planetary Science Letters*, 310(3-4): 286-292.
- Li, C., Lim, K., Berk, T., Abraham, A., Heisel, M., Guala, M., Coletti, F. and Hong, J., 2021. Settling and clustering of snow particles in atmospheric turbulence. *Journal of Fluid Mechanics*, 912: A49.
- Lube G, Breard ECP, Esposti-Ongaro T, Dufek J, Brand B. Multiphase flow behaviour and hazard prediction of pyroclastic density currents. *Nature Reviews Earth & Environment* (2020).
- Middleton, G.V. and Southard, J., 1984. *Mechanics of Sediment Movement*. Society of Economic Paleontologists and Mineralogists.
- Sulpizio, R., Bonasia, R., Dellino, P., Mele, D., Di Vito, M.A. and La Volpe, L., 2010. The Pomici di Avellino eruption of Somma–Vesuvius (3.9 ka BP). Part II: sedimentology and physical volcanology of pyroclastic density current deposits. *Bulletin of Volcanology*, 72(5): 559-577.
- Sulpizio, R. and Dellino, P., 2008. Sedimentology, Depositional Mechanisms and Pulsating Behaviour of Pyroclastic Density Currents. In: J. Gottsmann and J. Marti (Editors), *Caldera Volcanism: Analysis, Modelling and Response*. Developments in Volcanology. Elsevier, pp. 57-96.
- Sulpizio, R., Mele, D., Dellino, P. and La Volpe, L., 2007. Deposits and physical properties of pyroclastic density currents during complex Subplinian eruptions: the AD 472 (Pollena) eruption of Somma-Vesuvius, Italy. *Sedimentology*, 54(3): 607-635.
- Valentine, G.A., 1998. Damage to structures by pyroclastic flows and surges, inferred from nuclear weapons effects. *Journal of Volcanology and Geothermal Research*, 87(1-4): 117-140.
- Wang, L.-P. and Maxey, M.R., 1993. Settling velocity and concentration distribution of heavy particles in homogeneous isotropic turbulence. *Journal of Fluid Mechanics*, 256: 27-68.

REVIEWERS' COMMENTS

Reviewer #1 (Remarks to the Author):

I would like to thank the authors for their detailed responses to my comments and suggestions (and their similar attention to other reviewers). The quantitative descriptions of pressure fluctuations within pyroclastic density currents, and the relationship between those fluctuations and the turbulent structure of the currents are important for understanding the dynamics of these flows and their potential hazard. The paper is excellent and I look forward to seeing it published.
Benjamin Andrews

Reviewer #2 (Remarks to the Author):

After careful reading of the manuscript and reply to my comments I can recommend the manuscript for publication

Sincerely
Roberto Sulpizio

Reviewer #3 (Remarks to the Author):

The authors have responded well to my comments and have made relevant changes in the manuscript. The responses to the comments of the other reviewers seem to me to be convincing as well. This is a very interesting study. I recommend that the manuscript be accepted in present form.